# LAMBDA-SKIP CONNECTIONS: THE ARCHITECTURAL COMPONENT THAT PREVENTS RANK COLLAPSE

**Federico Arangath Joseph**
ETH Zurich
farangath@ethz.ch

**Jerome Sieber**
ETH Zurich
jsieber@ethz.ch

**Melanie N. Zeilinger**
ETH Zurich
mzeilinger@ethz.ch

**Carmen Amo Alonso**
Stanford University
camoalon@stanford.edu

## ABSTRACT

Rank collapse, a phenomenon where embedding vectors in sequence models rapidly converge to a uniform token or equilibrium state, has recently gained attention in the deep learning literature. This phenomenon leads to reduced expressivity and potential training instabilities due to vanishing gradients. Empirical evidence suggests that architectural components like skip connections, LayerNorm, and MultiLayer Perceptrons (MLPs) play critical roles in mitigating rank collapse. While this issue is well-documented for transformers, alternative sequence models, such as State Space Models (SSMs), which have recently gained prominence, have not been thoroughly examined for similar vulnerabilities. This paper extends the theory of rank collapse from transformers to SSMs using a unifying framework that captures both architectures. We study how a parametrized version of the classic skip connection component, which we call *lambda-skip connections*, provides guarantees for rank collapse prevention. Through analytical results, we present a sufficient condition to guarantee prevention of rank collapse across all the aforementioned architectures. We also study the necessity of this condition via ablation studies and analytical examples. To our knowledge, this is the first study that provides a general guarantee to prevent rank collapse, and that investigates rank collapse in the context of SSMs, offering valuable understanding for both theoreticians and practitioners. Finally, we validate our findings with experiments demonstrating the crucial role of architectural components such as skip connections and gating mechanisms in preventing rank collapse.

## 1 INTRODUCTION

The phenomenon of rank collapse has been recently reported in the deep learning literature Dong et al. (2023); Noci et al. (2022); Shi et al. (2022); He et al. (2023); Geshkovski et al. (2024); Wu et al. (2024a;b). Rank collapse is a phenomenon by which embedding vectors exhibit a fast convergence rate into a *uniform token*, or equilibrium embedding vector. Various explanations why this phenomenon arises have been proposed: Dong et al. (2023) proposed a path decomposition argument and suggests that one of the reasons on why rank collapse happens is that the attention matrix is row stochastic. Geshkovski et al. (2024) instead proposes an analysis of rank collapse from a statistical physics perspective by treating transformers as mean-field interacting particle systems and observes that the resulting system converges to a metastable state corresponding to a Dirac delta measure. In practice, this convergence to a uniform token reduces model expressivity Daneshmand et al. (2020); Dong et al. (2023); Noci et al. (2022); Wu et al. (2024b) since a low rank in the output matrix means that the model is unable to capture complex relationships between tokens, also potentially causing vanishing gradients during training (Noci et al., 2022). Therefore, gaining a deeper understanding of this phenomenon—its origins, conditions for occurrence, and potential prevention strategies—is essential for designing models that are more robust, stable, and expressive.

Rank collapse has primarily been studied in the transformer architecture Vaswani et al. (2023). In particular, it has been reported that the convergence rate to the equilibrium embedding vector of self-

attention Vaswani et al. (2023) (corresponding to a rank-1 matrix) is doubly exponential (Dong et al. (2023)). Both empirical observations and theoretical results support that the presence of skip connections (He et al. (2015)) as well as the LayerNorm component (Ba et al. (2016)) in the architecture mitigate the issue of rank collapse Dong et al. (2023); Wu et al. (2024a). In particular, Dong et al. (2023) shows that by adding skip connections, there exist an infinite amount of parametrizations of the network for which rank collapse does not happen. Wu et al. (2024a) instead argues that adding LayerNorm slows down rank collapse by showing that in this setting the upper bound on the rank collapse measure is higher than for networks without these components. This implies that, in order to have convergence to $0$, stricter conditions on input sequence and attention weights are needed.

Sequence models alternative to transformers have recently gained widespread attention. In particular, State Space Models (SSMs) such as S4 Gu et al. (2022b), S5 Smith et al. (2023), H3 Fu et al. (2023) and Mamba Dao and Gu (2024); Gu and Dao (2024) have been very prominent, where attention was replaced by SSM blocks, which operate in combination with other components such as Gating Mechanisms (Hochreiter and Schmidhuber (1997); Chung et al. (2014)), MLPs (Popescu et al. (2009) and skip connections (He et al. (2015)). Furthermore, comparisons between these architectures have been discussed in (Sieber et al., 2024; Hè and Kabic, 2023; Tiezzi et al., 2024), and a larger framework where both the transformer and SSMs are seen as different realizations of the same general model has recently been proposed (Ali et al. (2024), Dao and Gu (2024)). However, whether they suffer from similar weaknesses, and whether rank collapse happens more broadly in these architectures, is still an open question.

In this work, we focus on the specific issue of rank collapse and explore how slight modifications in the architecture, referred to as lambda-skip connections, can help to prevent it. Although parametrized versions of skip connections have been explored in the past (see e.g. He et al. (2016),Srivastava et al. (2015)), their impact in providing guarantees to prevent rank collapse has not been studied. Moreover, here we develop a general theory of rank collapse using the unifying framework for sequence models presented in Ali et al. (2024) and (Dao and Gu, 2024), comprising both transformers and SSM architectures. To the best of our knowledge, the rank collapse phenomenon in the SSM setting has not yet been studied and we are the first to provide a general lower bound for rank collapse which holds for both Transformers and SSMs. Additionally, we offer a mechanistic solution to prevent rank collapse by introducing a skip strength parameter, which comes with minimal computational overhead while enhancing the model's expressivity and stability. Since rank collapse can be a very serious issue for practitioners, investigating broadly when rank collapse occurs and designing robust architectures that prevent rank collapse is of central interest not only to theorists but also to experimentalists. Our contributions can be outlined as follows:

- We extend the theory on rank collapse originally developed for transfomers to SSMs by using the recently proposed framework unifying these two architecture types (Ali et al., 2024; Dao and Gu, 2024).

- We investigate how lambda-skip connections influence rank collapse. In particular, we give sufficient conditions that guarantee that rank collapse does not occur in the finite layers setting. Our conditions apply to both transformers and SSMs. We also study the necessity of this condition via ablation analysis and analytical examples.

- We provide experimental details in support of our theory, by showing how adding a parameter controlling the skip connection strength could be beneficial in preventing rank collapse.

- We also empirically show the role of gating mechanisms in preventing rank collapse. To the best of our knowledge, we are the first to make this connection between gating mechanisms, which were originally constructed for improving memory capabilities of the models (Hochreiter and Schmidhuber, 1997), and rank collapse.

## 2 RELATED WORKS

**Rank Collapse.** Rank collapse is the phenomenon in which the rank of the output layer of the network collapses to 1 (Dong et al. (2023)). To mathematically capture this, measuring the rank is not ideal since for every finite amount of layers, the output matrix will always be full rank with probability 1. This led to the development of an alternative metric to measure rank collapse Dong et al. (2023), which intuitively evaluates how "ill-conditioned" is the matrix of the output layer, i.e.,

how close it is to being rank 1. In Dong et al. (2023), this metric is defined by computing how far the columns of the matrix are from the mean of all columns. It was shown in e.g. Dong et al. (2023); Wu et al. (2024a) that this metric quickly decays to 0 as the depth of the model increases. Hence, rank collapse is an intrinsic issue of deep-learning models. Rank collapse has mainly been studied for transformers (Vaswani et al. (2023)), Graph Neural Networks (Zhou et al. (2021)) and ReLU networks (Agarap (2019)), but not SSMs. The work in Daneshmand et al. (2020) studied rank collapse in randomly initialized linear and ReLU networks and proved that Batch Normalization can be helpful to mitigate the phenomenon. In Dong et al. (2023) the authors show that in self-attention-only transformers, rank collapse occurs doubly exponentially, i.e. with rate $O(a^{b^n})$ for some $a, b \in \mathbb{R}$. In the same paper, the authors propose for the first time that skip connections and MLPs (Popescu et al. (2009)) are helpful in preventing rank collapse from happening. Moreover, the work in Noci et al. (2022) shows that rank collapse is not only an issue during inference but also hinders training due to vanishing gradient problems at initialization. In Wu et al. (2024b) the authors show that rank collapse also happens in graph attention networks (in this setting the phenomenon is also known as oversmoothing), also causing the model to lose expressive power. Finally, the work in Wu et al. (2024a) studies the effect of LayerNorm on rank collapse in transformers, by proving that this component can help preventing or mitigating rank collapse.

**Skip Connections.** Skip connections were first introduced in ResNets He et al. (2015) with the purpose of easing training and optimization in deep neural networks (He et al. (2016); Veit et al. (2016); Balduzzi et al. (2018). More precisely, the introduction of skip connections addresses the vanishing gradient problem by decomposing back-propagation of gradients into multiple paths and thus allowing them to skip and bypass layers. As an alternative explanation, it was shown in Li et al. (2018) that adding skip connections also results in a loss function which is much smoother and hence it is harder for gradients to get stuck in local minima, facilitating the training procedure. In transformers and SSMs, the introduction of the skip connection followed the same motivation as for ResNets and other deep neural networks, i.e. to improve and facilitate optimization. The work in Dong et al. (2023) is the first to also connect the benefit of skip connections to avoiding rank collapse by studying the paths decomposition in transformers. Regarding modifications to the skip connection, Bachlechner et al. (2020) also considers a control parameter in residual networks. However, their approach differs in that it is used as a gating with the main network layer again with the purpose of stabilizing optimization, which allows for training of much deeper models. The work in Srivastava et al. (2015) proposes adding an extra feedforward layer to each residual layer. The output of each residual layer is then expressed as a convex combination of two terms: one gated by the residual connection and the other by the output of the primary network layer. The effect of these modifications on skip connection have not been analyzed. Lastly, He et al. (2016) studies the lambda-skip connection proposed here in terms of training efficiency and concludes that the standard choice of $\lambda = 1$ is usually beneficial.

**LayerNorm.** LayerNorm (Ba et al. (2016)) has become the standard choice of normalization layer for sequence models such as transformers (Vaswani et al. (2023)) and Mamba (Dao and Gu (2024)). In contrast to batch normalization (Ioffe and Szegedy (2015)), LayerNorm performs the shifting and the normalization token-wise and not sequence-wise. This is particularly beneficial as the embedding dimension is usually much smaller than the sequence dimension, resulting in a much cheaper computational overhead than BatchNorm. It has also been shown that LayerNorm is helpful to stabilize training (Xiong et al. (2020)) and to increase the expressivity of the attention layer (Brody et al. (2023)). Related instead to rank collapse, the work in Dong et al. (2023) was the first to analyze a potential connection between rank collapse and LayerNorm. However, they conjecture that LayerNorm has no impact on rank collapse. Later on, the work in Wu et al. (2024a) refutes this claim, showing both empirically and theoretically that LayerNorm helps slow down rank collapse.

## 3 PROBLEM SETUP

### 3.1 SEQUENCE MODEL DEEP-LEARNING ARCHITECTURE

Rank collapse is defined as the convergence – in the limit of infinite depth – of the output matrix of the network's layers to a Rank-1 matrix. In this sense, rank collapse is intrinsic to deep-learning architectures. In what follows, we outline the different components on a *single-layer* of the architecture considered in this paper. We note that the presented architecture is general enough to capture both transformers and SSM-like architectures.

We start by introducing some notation. We index each layer of the architecture with $k \in \{1, ..., K\}$. We denote $X^{(k)} \in \mathbb{R}^{N \times d}$ and $Y^{(k)} \in \mathbb{R}^{N \times d}$ to be the input and output of the $k$-th layer, respectively, where $N$ is the sequence length. In this setting, $Y^{(0)}$ is the input sequence to the model. Each layer $k$ has the following components.

**Main mechanism.** The main mechanism differs between attention in transformers, and a recurrent block in SSMs. However, as shown in Ali et al. (2024) and Dao and Gu (2024), both models can be seen as part of a general framework

$$O^{(k)} = M^{(k)} V^{(k)}. \tag{1}$$

In the case of attention,

$$V^{(k)} = X^{(k)} W_V^{(k)} \in \mathbb{R}^{N \times d} \quad \text{and} \quad M^{(k)} = \text{softmax}\left(\frac{X^{(k)} W_Q^{(k)} (W_K^{(k)})^T (X^{(k)})^T}{\sqrt{d_{QK}}}\right) \in \mathbb{R}^{N \times N},$$

with $W_Q^{(k)} \in \mathbb{R}^{d \times d}$, $W_K^{(k)} \in \mathbb{R}^{d \times d}$ and $W_V^{(k)} \in \mathbb{R}^{d \times d}$ are the query, key and value matrices respectively.

In the case of the recurrent block, $V^{(k)} = X^{(k)}$, and $M^{(k)}$ is a lower triangular matrix with

$$M_{ji}^{(k)} = C_j^{(k)} \left(\prod_{l=0}^{j-i-1} A_{j-l}^{(k)}\right) B_i^{(k)},$$

where $A_t^{(k)} \in \mathbb{R}^{dH \times dH}, B_t^{(k)} \in \mathbb{R}^{dH \times 1}, C_t^{(k)} \in \mathbb{R}^{1 \times dH}, U_t^{(k)} \in \mathbb{R}^{1 \times 1}$ (usually it is indicated with $D_t^{(k)}$, but we decide to use a different notion to avoid confusion with LayerNorm). The first versions of the SSM blocks consisted of Linear Time Invariant (LTI) representations, i.e., $A_t^{(k)} = A^{(k)}$, $B_t^{(k)} = B^{(k)}$, $C_t^{(k)} = C^{(k)}$ and $U_t^{(k)} = U^{(k)}$ for all time steps $t$ (Gu et al., 2022b;a; Smith et al., 2023). Recently, in order to improve the expressivity and the performance of such models, selective State Space Models have been proposed (Gu and Dao, 2024; Dao and Gu, 2024). In these models, matrices $A_t$, $B_t$, and $C_t$ are input-dependent. We refer to Appendix A.1 for a detailed explanation of attention and the recurrent block.

**Skip Connection.** The skip connection sums the unmodified input to the output of the self-attention layer:

$$\tilde{Y}^{(k)} = X^{(k)} + O^{(k)}. \tag{2}$$

Although it was originally proposed to stabilize training by avoiding vanishing gradients (He et al. (2015)), it was recently shown that it also has a role in modulating rank collapse Dong et al. (2023). In SSM architectures, skip connections are often preceded by gating mechanisms, i.e., $O^{(k)}$ in equation 2 is replaced by $\tilde{O}^{(k)} = O^{(k)} \odot \sigma(W^{(k)} X^{(k)})$.[1]. However, we ignore these in the theoretical part of this paper for simplicity. In this work, we modify the skip connection by allowing it to have an additional parameter $\lambda^{(k)} \in \mathbb{R}$, that controls the strength of the skip connection, i.e.,

**Definition 3.1.** *Given a standard skip connection as per equation 2 and a parameter $\lambda^{(k)} \in \mathbb{R}$, a* lambda-skip connection *is defined as*

$$\tilde{Y}^{(k)} = \lambda^{(k)} X^{(k)} + O^{(k)}. \tag{3}$$

In the theoretical section, we will consider $\lambda^{(k)}$ to be fixed for all the layers for simplicity. However, in the experimental section, we will also consider models with learnable $\lambda^{(k)}$, i.e. it might vary across layers.

**Layer Norm.** The LayerNorm (Ba et al., 2016) computation shifts and normalizes the output $\tilde{Y}^{(k)}$ after the skip connection. Similar to the skip connection, the purpose of the LayerNorm is also to stabilize training. It was recently shown to also help to mitigate rank collapse (Wu et al., 2024a). Here, we consider a slightly simplified version of LayerNorm, similar to Wu et al. (2024a), where we only apply normalization and not shifting, namely:

$$Y^{(k)} = D^{(k)} \tilde{Y}^{(k)}, \tag{4}$$

where $D^{(k)} = \text{diag}(d_1^{(k)}, d_2^{(k)}, ..., d_N^{(k)}) \in \mathbb{R}^{N \times N}$ and $d_i^{(k)} = \frac{1}{||\tilde{Y}_{i,:}^{(k)}||_2}$.

---

[1] $\sigma(\cdot)$ is a non-linearity, often SILU, $W^{(k)}$ is a weight matrix and $\odot$ is the Hadamard product.

**Other components.** After the above-mentioned computations, the input $Y$ passes through an MLP (Popescu et al., 2009), which can intuitively be seen as a non-linear feature extractor or a kernel. MLPs might play a role in mitigating rank collapse since they increase the Lipschitz constant of the network (Dong et al., 2023). In particular, the Lipschitz constant of the MLP layer appears in the double exponential decay term of the upper bounds of the rank collapse measure. This is beneficial since it allows for a wider range of choices for the parametrization of other components while maintaining a high-enough upper bound. Since the role of the MLP layer on rank collapse is well-understood, in this work we focus on the effect that LayerNorm and skip connections have on rank collapse. We anticipate that the addition of MLP will further improve the bound, and hence the rank collapse avoidance conditions.

## 3.2 PROBLEM STATEMENT

In mathematical terms, the phenomenon of rank collapse means that the columns of the final output matrix are proportional to each other. The representational capacity of the model is severely hindered by this phenomenon: since the final representations of each token are proportional to each other, there is limited diversity of tokens to be chosen from. Given that the rank of the output layer will be full rank with probability one (since rank-deficient matrices are a zero measure subspace of the space of all matrices), a different metric was introduced in Dong et al. (2023); Wu et al. (2024a) to assess rank collapse:

$$\mu(Y^{(k)}) = ||Y^{(k)} - \mathbf{1}\gamma_{Y^{(k)}}||_F, \tag{5}$$

where $Y^{(k)} \in \mathbb{R}^{N \times d}$ is the output matrix resulting from the k-th layer, $\gamma_{Y^{(k)}} := \frac{1}{N}\mathbf{1}^T Y^{(k)}$, and $||\cdot||_F$ is the Frobenius norm.

In order to study the impact of the architecture on the phenomenon of rank collapse, we make use of the equations describing the architecture: equation 1, equation 3, and equation 4. Moreover, we note that the output of layer $k-1$ is the input to layer $k$, i.e. $X^{(k+1)} = Y^{(k)}$. Combining these equations we obtain

$$Y^{(k)} = D^{(k)}(M^{(k-1)}Y^{(k-1)}C_V^{(k-1)} + \lambda^{(k)}Y^{(k-1)}), \tag{6}$$

where $C_V^{(k)} = W_V^{(k)}$ for the attention block and $C_V^{(k)} = I$ for the recurrent block.

The goal of this paper is to show how the introduction of $\lambda^{(k)}$ in the lambda-skip connection influences rank collapse. For this purpose, the main quantity of interest is $\mu(Y^{(K)})$ i.e., the value of the token-similarity metric on the output of the last layer. We provide a mathematical analysis of this metric in terms of the different architectural components in the next section.

## 4 LAMBDA-SKIP CONNECTION: NECESSARY AND SUFFICIENT TO PREVENT RANK COLLAPSE

Here, we present the analysis resulting from studying rank collapse with lambda-skip connections. In Section 4.1, we analyze the scenario where we replace the regular skip connection by a lambda-skip connection and show that, together with LayerNorm, an appropriate choice of $\lambda$ in equation 3 is sufficient to prevent rank collapse in the finite layers setting. For simplicity, here we consider a fixed value of $\lambda$ for all the layers, i.e. $\lambda^{(k)} = \lambda \ \forall k$. Specifically, we provide the value of $\lambda \in \mathbb{R}$ in the lambda-skip connection that guarantees the absence of rank collapse. This result holds for any arbitrary amount of layers and all the three architectures studied in the paper (transformers, LTI SSMs and selective SSMs). In Section 4.2, we study the necessity of this condition. First we show that in the ablated architecture without skip connection, rank collapse occurs exponentially, i.e., $\mu(Y^{(K)})$ converges to 0 exponentially. If, additionally, LayerNorm is also ablated, rank collapse occurs doubly exponentially. Second, we highlight via examples that the strength of the lambda-skip connection (parametrized through $\lambda \in \mathbb{R}$) plays a crucial role in determining the effect of skip connections on rank collapse.

## 4.1 LAMBDA-SKIP CONNECTION: SUFFICIENT TO PREVENT RANK COLLAPSE

Here we show the role that the parameter $\lambda \in \mathbb{R}$ plays in guaranteeing, under appropriate conditions, the prevention of rank collapse. To do this, we provide a $\lambda$-dependent lower bound for the rank

collapse metric $\mu(Y^{(K)})$ in equation 5. We start by defining the following quantity:

$$b := \frac{1}{a^K} \frac{2\lambda N d S C_M}{\lambda^2 - a(SC_M + |\lambda|)^2},$$

with $C_M := \sup_k ||M^{(k)}||_F$, $S := \sup_k ||C_V^{(k)}||_F$. We also define the collapse rate as:

**Definition 4.1.** *The* collapse rate*, $a > 0$, is the rate at which the lower bound for the rank collapse metric $\mu(Y^{(K)})^2$ decays with the number of layers, i.e $\mu(Y^{(K)})^2 \geq a^K \mu(Y^{(0)})^2$.*

**Theorem 4.1** (Lower Bound on Rank Collapse). *Let the input sequence $Y^{(0)}$ be such that $\mu(Y^{(0)})^2 \geq b$. If the skip connection strength $\lambda$ is chosen to satisfy*

$$\lambda^2 - a(SC_M + |\lambda|)^2 > 0, \tag{7}$$

*then we can lower bound $\mu(Y^{(K)})$ by $\mu(Y^{(K)})^2 \geq a^K \mu(Y^{(0)})^2$ for all $K \in \mathbb{N}$.*

*Proof.* The key part of the proof is to show the following relationship between the rank collapse measure at two consecutive layers: $\mu(Y^{(k+1)})^2 \geq \frac{1}{(SC_M + |\lambda|)^2} \left( \lambda^2 \mu(Y^{(k)})^2 - 2\lambda N d C_M \right)$, which is done by algebraic manipulation and upper bounds on $||Y^{(k+1)}||_F$. The claim follows by imposing the right-hand side above greater of equal to $a\mu(Y^{(k)})^2$, rearranging the terms and imposing the condition on $\lambda$ to guarantee the feasibility of this. The full proof can be found in Appendix A.4. $\square$

We note that this result is very general, since it holds for any model that can be expressed in the form of equation 6, independent of its specific implementation or parametrization. Moreover, the presence of LayerNorm implies that the bound provided in equation 7 does not depend on the input even when the matrix $M$ is input-dependent. This is because the input is normalized before being fed through $M$. Specifically, $C_M$ is only dependent on the weights and on the sequence length [2]

Theorem 4.1 also provides an important intuition for the choice of $\lambda$. In particular, we note that to guarantee prevention of rank collapse, the ideal choice is $a = 1$. However, this choice will be mediated by the values of $C_M$ and $S$ in the different architectures. In particular, it is easy to see that the only way to guarantee a solution to 7 is by having $1 - a > 0$, for which $\lambda$ needs to satisfy $|\lambda| > \frac{(a + \sqrt{a})SC_M}{1-a}$ .

**Remark 4.1.** *In the case of Mamba, $C_V = I$ and $c = S = 1$. This implies that the only possible choice for the collapse rate is $a < 1$. Although this does not guarantee $\mu(Y^{(K)})^2 \geq \mu(Y^{(0)})^2$, in practice $a$ can be chosen to be very close to 1. For instance, one can set $a = 0.9999$. Given $K = 64$ (standard number of layers in Mamba), $a^K \approx 0.993$. Hence, in practice, if we choose $\lambda$ appropriately, this choice still prevents rank collapse (see e.g. Figure 2).*

Finally, we conclude this section by briefly discussing how Theorem 4.1 can be generalized to the case where we let $\lambda$ vary across different layers. The following small modification will be enough to make the result still hold in this setting: in order to still satisfy the condition $\mu(Y^{(k)}) \geq a\mu(Y^{(k-1)})$, we can simply adapt equation 7 to be $\lambda_k^2 - a(S_k C_{M_k} + |\lambda_k|)^2 > 0$, where $S_k = ||C_V^{(k)}||_F$ and $C_{M_k} = ||M^{(k)}||_F$. Additionally, the flexibility to choose $\lambda_k$ allows for different values of $a$ to be used across layers. For instance, instead of setting a single $a$ value for all layers, one can vary $\lambda_k$ to meet different layer-specific conditions. In one layer, $\lambda_k$ might satisfy the condition for a smaller value of $a$, while in a subsequent layer, $\lambda_k$ satisfies the condition for a larger value of $a$.

## 4.2 LAMBDA-SKIP CONNECTION: NECESSARY TO PREVENT RANK COLLAPSE?

Next, we explore the role that the lambda-skip connection plays in terms of being necessary to prevent rank collapse. Although we do not provide a formal necessary condition, we explore this idea in two ways. First, we provide theoretical results on how rank collapse occurs when the (standard) skip connection is ablated. Then, we give two examples of how rank collapse may occur when the lambda-skip connection does not attain the bound in Theorem 4.1.

---

[2]In transformers $C_M = \sqrt{N}$, in structured LTI SSMs $C_M \geq \sup_k ||A^{(k)}|| \, ||B^{(k)}|| \, ||C^{(k)}||_F$ and in selective SSMs $C_M \geq \sup_k ||W_B^{(k)}|| \, ||W_C^{(k)}||_F$ (since in Selective SSMs it is common practice to choose $A_t^{(k)}$ with eigenvalues smaller than one).

### 4.2.1 RANK COLLAPSE OCCURS WITHOUT SKIP CONNECTIONS

Here we show how, for different architectures, rank collapse occurs in the absence of skip connections. Here we focus on transformers and selective SSMs. Additional results on LTI SSMs are presented in Appendix A.8.

**Transformers.** We start by reporting for completeness the result in Wu et al. (2024a) for transformers architectures with Self-Attention and LayerNorm only.

**Theorem 4.2** (Corollary 1, Wu et al. (2024a) (Informal))**.** *Under certain assumptions on the self-attention weights and the input sequence, there exist $C > 0$, $\epsilon > 0$ and $r > 0$ such that $N\epsilon < 1$ and*

$$\mu(Y^{(K)}) \leq C(1 - \epsilon^{2r})^{\frac{K}{2r}}, \quad \forall K \geq 0.$$

*meaning that tokens converge to a common point on $\mathbb{S}^{d-1}$ exponentially.*

We refer to Appendix A.5 for a rigorous presentation of Theorem 4.2. The main takeaway from this result is that, under appropriate conditions on the architecture and the input sequence, removing skip connections and only using attention layers and LayerNorm can cause an exponentially fast convergence to a rank-1 matrix. This highlights the importance of skip connections to mitigate rank collapse. It is also interesting to compare the result in Theorem 4.2 to the setting of self-attention only transformers (without both LayerNorm and skip connections), reported in Dong et al. (2023). Under certain assumptions on the attention matrix and the input sequence, rank collapse occurs at a doubly exponential rate. We report the full result in Theorem A.3 and Corollary A.3.1 (where we adapted the Theorem to the rank collapse measure provided in the problem statement).

**Selective SSMs.** According to Dao and Gu (2024), for selective SSMs where $A_t = \alpha_t I$ (e.g., Mamba-2), the matrix $M$ can be compactly written as $M^{(k)} = 1\text{SS}(\alpha) \odot \left( Y^{(k)} W_C W_B^\top Y^{(k)T} \right)$, where $1\text{SS}(\alpha)$ is a lower-triangular 1-semiseparable matrix. Due to space limitations, we formally define $1\text{SS}(\alpha)$ in Equation 12 in Appendix A.3. In what follows, we introduce the following assumption on selectivity:

**Assumption 4.1.** *Only the matrices $B_t = Y_{t,:}^{(k)} W_B$ and $C_t = Y_{t,:}^{(k)} W_C$ are input-dependent, while $A_t = A = \alpha I \ \forall t$ (with $\alpha \leq 1$) is independent of the input.*

This choice is done for ease of exposition. Although it may seem restrictive in theory, we show experimentally that eliminating skip connections can cause rank collapse for selective SSMs where $A_t$ is input dependent as well, such as Mamba (see e.g. the plot corresponding to $\lambda = 0$ in Figure 1). In the following derivation, we neglect gating mechanisms for the sake of the theoretical analysis–we will thoroughly analyze their effect on rank collapse in simulation (see e.g. Figure 3).

**Theorem 4.3** (Rank Collapse for selective SSMs without skip connection)**.** *Let $\phi^{(k)} = \min_{i,j \in [N]} \langle Y_{i,:}^{(k)}, Y_{j,:}^{(k)} \rangle$, where $\langle \cdot, \cdot \rangle$ indicates the inner product. Under Assumption 4.1 and if $c \leq \phi^{(0)} < 1$ for some $c > 0 \in \mathbb{R}$, $\lambda_{min} > 0$, $\sum_j M_{ij} \geq 1 \ \forall i$, then it holds that:*

$$\mu(Y^{(K)}) \leq \sqrt{N} \left( 1 - c^2 \lambda_{min}^2 \alpha^{2N} \right)^K \ \forall K \geq 0,$$

*where $\lambda_{min}$ and $\lambda_{max}$ the minimum and maximum eigenvalues of $\frac{W_B W_C^\top + W_C W_B^\top}{2}$.*

*Proof.* The proof adapts the derivation in Corollary 1 in Wu et al. (2024a) to selective SSMs. In particular, the key is to find a recursive relation between $1 - \phi^{(k)}$ and $1 - \phi^{(k+1)}$. The full proof can be found in Appendix A.7. □

In particular, when $\lambda_{\min} \leq \frac{2}{c^2 \alpha^{2N}}$, the model suffers from exponential rank collapse. Note that the above result holds for a model with layers having the same parametrization.

In Appendix A.9 Theorem A.10, we show that if we instead have a selective SSM with ablated the skip connections and LayerNorms, then rank collapse can happen doubly exponentially under certain conditions, similarly to what we have in transformers architecture. In contrast with the LTI SSM case (Theorems A.7 and A.8), which exhibits an exponential decay, the matrix $M^{(k)}$ depends quadratically on the input for selective SSMs: $||M^{(k)}||_F \leq \sqrt{N} ||Y^{(k)}||_F^2 ||W_{BC}||_F$. where $W_{BC} = W_C W_B^\top$. This dependency is what causes the doubly exponential behavior.

### 4.2.2 RANK COLLAPSE CAN STILL OCCUR WITH SKIP CONNECTIONS UNDER SPECIFIC CHOICES OF $\lambda$: EXAMPLES

In the following, we show an analytical example of how, for Selective SSMs, ank collapse can still occur in the presence of lambda-skip connections for specific choice of $\lambda$. A similar example for Structured LTI SSMs can be found in Appendix A.10 .

**Selective SSMs.** Consider the system, which we will denote as [Sys-2], $A_t = A = \alpha I$ (for simplicity, we consider $\alpha$ not to be input dependent here), $B = YW_B$ and $C = YW_C$ input dependent (where $y_t$ is the $t$-th row of $Y$) with LayerNorm applied to the output of each layer. Once again, we let $N = 2$ and $d = 2$. We have $M = W_A \odot \left( YW_C W_B^\top Y^\top \right)$ where $W_A = \begin{pmatrix} 1 & 0 \\ \alpha & 1 \end{pmatrix}$. By choosing $\alpha = 1$ and $W_B = W_C = I$, we get that $M = \begin{pmatrix} 1 & 0 \\ y_1^\top y_2 & 1 \end{pmatrix}$ since the input to the layer is normalized by LayerNorm.

**Proposition 4.3.1.** *Given [Sys-2], if we then choose $\lambda > -\frac{3}{2}$, $M$ as above and $Y^{(0)} = \begin{pmatrix} 1 & 0 \\ \frac{\alpha_0}{\sqrt{\alpha_0^2 + \beta_0^2}} & \frac{\beta_0}{\sqrt{\alpha_0^2 + \beta_0^2}} \end{pmatrix}$ for any $\alpha_0, \beta_0$ with $\alpha_0 > 0$, then we have that $\mu(Y^{(k)}) \to 0$. On the other hand, if we choose $\lambda < -\frac{3}{2}$ and $M$ and $Y^{(0)}$ as above, then rank collapse is avoided.*

*Proof.* Again, the result follows by calculating $Y^{(k)}$ inductively and by showing that the corresponding $\mu(Y^{(k)})$ decays to 0 for $\lambda > -\frac{3}{2}$ while this does not happen if $\lambda < -\frac{3}{2}$. We refer to Appendix A.11 for the full proof. $\qquad\square$

### 4.2.3 TIGHTNESS OF LOWER BOUND IN THEOREM 4.1

We conclude this section with a discussion of the tightness of the lower bound provided in Theorem 4.1. In particular, we explore whether there exist a system such that $\mu(Y^{(k)})^2 = O\left(a^k \mu(Y^{(0)})^2\right)$ for a choice of $\lambda$ that satisfies Equation 7. We show in Proposition 4.3.2 that such a system exists. Hence, without additional assumptions on the architecture or on the input sequence, it is not possible to provide a better guarantee for the lower bound of the rank collapse metric.

**Proposition 4.3.2.** *There exist an architecture of the form of Equation 6 such that, when choosing $\lambda$ to satisfy Equation 7, i.e. $|\lambda| = \Omega(\frac{a}{1-a})$, it holds that $\mu(Y^{(k)})^2 = O\left(a^k \mu(Y^{(0)})^2\right)$.*

*Proof.* We again consider [Sys-2] from Proposition 4.3.1 with $\alpha_0 = \beta_0 = \frac{1}{2}$. From the proof of Proposition 4.3.1, we calculate the values of $Y^{(k)}$ and $\mu(Y^{(k)})$ for all layers $k$. Then, we calculate the value of $a$ by simply considering the ratio of the rank collapse measure at two consecutive layers. Then, the result simply follows by upper bounding the value of $a$ and comparing it with the desired bound. The full proof can be found in Appendix A.12. $\qquad\square$

## 5 EXPERIMENTS

In this section, we aim to both explore the link between gating mechanisms and rank collapse and to empirically validate our theoretical findings. In Section 5.1, we empirically validate Theorem 4.1, demonstrating the importance of selecting the appropriate skip connection strength to mitigate rank collapse. In Section 5.2, we show that for the Mamba-2 architecture (Dao and Gu (2024)), gating mechanisms indeed play a crucial role in preventing rank collapse. Finally, we provide additional experiments on the S4 architecture (Gu et al., 2022b) in Appendix A.13.1.

### 5.1 COMPARISON OF RANK COLLAPSE FOR MAMBA ARCHITECTURE WITH DIFFERENT VALUES OF $\lambda$

We start by validating our main finding, i.e. that by controlling the value of $\lambda$ in the skip connection, we can prevent rank collapse from happening. In order to do so, we consider a 2 billion parameters pre-trained Mamba-2 model available on HuggingFace and remove the gating mechanisms. Instead, we add an additive skip connection with varying $\lambda$. For simplicity, we keep the value of $\lambda$ constant at every layer. Similar to Dong et al. (2023); Wu et al. (2024a), we sample 32 text excerpts from

Wikipedia using the Wikipedia API and tokenize them in sequences of at least 128 tokens using the BERT tokenizer Devlin et al. (2019). We then forward pass these inputs through the pre-trained model and evaluate the rank collapse measure $\mu$ (normalized by the norm of the layer output) for each of the 64 layers for different values of $\lambda$, as shown in Figure 1. We observe that when $|\lambda|$ is small, the model suffers from rank collapse. This happens even when $\lambda = 1$, which is usually the default design choice. After sufficiently increasing or decreasing $\lambda$, we note that we avoid rank collapse, consistent with our theoretical results, particularly Theorem 4.1. This suggests that rather than fixing the skip connection strength at $\lambda = 1$ (as is standard), it may be beneficial to treat $\lambda$ as a learnable parameter or hyperparameter to improve model stability. In Figure 2, we analyze

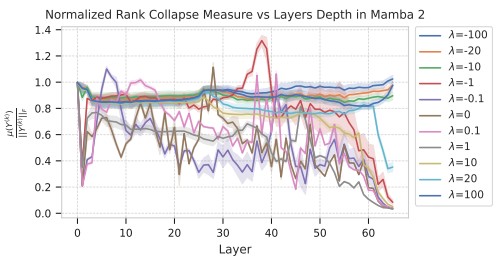

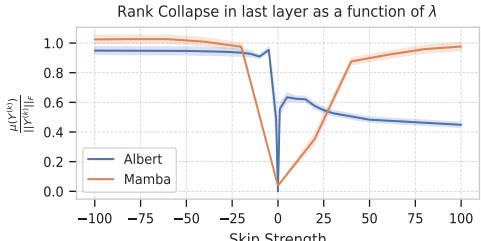

Figure 1: (Normalized) Rank Collapse measure plotted for different values of the skip connection's strength $\lambda$. Again, shaded areas represent one standard deviation from the mean calculated over the 32 examples.

Figure 2: (Normalized) Rank Collapse measure at the last layer plotted for different values of $\lambda$ for both Albert and Mamba. Shaded areas represent one standard deviation from the mean calculated over the 32 examples.

how the rank collapse measure at the last layer varies with different skip connection strengths for both the Albert's architecture (Lan et al. (2020)) and Mamba-2, keeping the full architecture (also LayerNorm). In both cases, we note the importance of tuning and choosing the correct value of $\lambda$ to avoid rank collapse. We see how a value of $\lambda$ very close to 0, as we expected, is detrimental and causes a very low value of $\mu$, even in the presence of LayerNorm, which is consistent with our findings in Theorem 4.3. Additionally, we see that a negative $\lambda$ leads to a higher $\mu$. Intuitively, the reason for this could be that a negative $\lambda$ can be interpreted as negative feedback in the system, hence helping stabilizing it. Note that even if in both cases our condition on $\lambda$ in Theorem 4.1 is too conservative, in practice much lower values of $\lambda$ are good enough to prevent rank collapse. In Figure 2 we note that as $|\lambda|$ increases, there is little variation in $\mu$. This can be explained using the lower bound proposed in Theorem 4.1. Specifically, as $|\lambda| \to \infty$, the parameter $a \to 1$. This implies that the rank collapse metric decreases much more slowly with the number of layers. In the limit, it does not decrease at all, remaining at or above the rank collapse measure at the input.

We investigate whether using $\lambda$ as a trainable parameter affects the learning performance. To this end, we train two transformer-based architectures and two SSM-based architectures on the image task of the long range arena (LRA) benchmark (Tay et al., 2020) and on a multi-query associative recall (MQAR) task (Arora et al., 2023). The accuracy of all four architectures on both tasks are reported in Table 1, and the details on the experimental setup are provided in Appendix A.14. From our observations, we conclude that learning $\lambda$ does not affect the performance and even outperforms the models with fixed $\lambda$ in some cases.

| Task [%] | Model | | | | | | | |
|---|---|---|---|---|---|---|---|---|
| | Transformer | | Lin. Transformer | | Mamba | | Mamba-2 | |
| | $\lambda = 1$ | var. $\lambda$ | $\lambda = 1$ | var. $\lambda$ | $\lambda = 1$ | var. $\lambda$ | $\lambda = 1$ | var. $\lambda$ |
| Image (LRA) | 32.64 | 32.85 | 34.10 | 32.80 | 63.04 | 62.92 | 42.28 | 38.92 |
| MQAR ($L = 512$) | 99.6 | 98.9 | 1.6 | 1.55 | 81.5 | 85.3 | 97.3 | 99.1 |

Table 1: Model performance in terms of test accuracy on the `Image` task of the LRA benchmark and the MQAR task $\{L = 512, \text{KV-pairs} = 64\}$.

## 5.2 COMPARISON OF RANK COLLAPSE FOR MAMBA ARCHITECTURE WITH AND WITHOUT GATING MECHANISMS AND LAYERNORM

We also analyze the effect of gating mechanisms and LayerNorms in the Mamba architecture. Because gating mechanisms can be viewed as a multiplicative form of skip connections, we aim to empirically explore whether they, like skip connections, contribute to stability and prevent rank collapse. We repeat the same procedure as above, i.e., forward passing 32 excerpts from Wikipedia to the pre-trained model and calculating the rank collapse measure across all layers. We replicate this for all considered cases, namely gating plus LayerNorm, only gating, only LayerNorm, and neither gating nor LayerNorm.

As shown in Figure 3, when removing the gating mechanism, we either observe the rank collapse measure approaching zero especially for later layers if the LayerNorm is present, or we observe an initial drop in the rank collapse measure followed by a dramatic increase if we remove the LayerNorm. This highlights the importance of having LayerNorm to keep the model stable, as was also observed in Dao and Gu (2024). Indeed, even when the model contains a gating mechanism, although we prevent rank collapse, removing LayerNorm leads to a sudden increase in the rank collapse measure, indicating instability. Interestingly, we find that gating mechanisms, originally designed to enhance memory capabilities, also play a crucial role in preventing rank collapse. To the best of our knowledge, this is the first time this connection has been made.

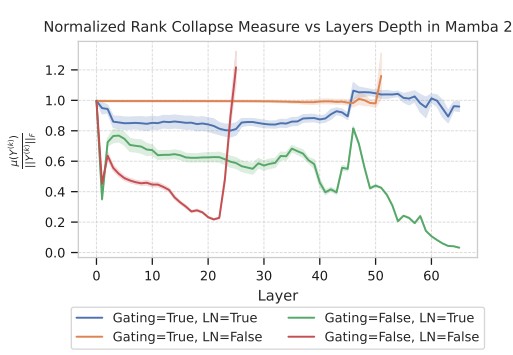

**Figure 3:** (Normalized) Rank Collapse measure of the Mamba-2 model plotted as a function of layer depth. The shaded areas represent one standard deviation from the mean calculated over the 32 examples. Gating=True/False indicates we use a Mamba2 architecture with/without gating whereas LN=True/False indicate we use the architecture with/without LayerNorm

## 6 CONCLUSION

This paper studies the phenomenon of rank collapse across general sequence model architectures, using the unifying framework from Ali et al. (2024) and Dao and Gu (2024). Our main contribution is the introduction of a lambda-skip connection, which adds a regulatory term to the standard skip connection component to control its strength. We provide sufficient conditions for this parameter under which rank collapse does not occur. We do this via a lower bound on the rank collapse measure, which holds for transformers, LTI SSMs, selective SSMs and more generally, all architectures that can be written in the framework from Ali et al. (2024) and Dao and Gu (2024). We further study the necessity of the lambda-skip connection by showing that when this component is ablated, selective SSMs also experience rank collapse even in the presence of LayerNorm. Moreover, we show that similarly to Transformers, selective SSMs can also be affected by doubly exponential rank collapse when both skip connections and LayerNorm are ablated. We suggest that the primary cause of this doubly exponential rank collapse is the input dependence of the $M^{(k)}$ matrix. We validate our theoretical analysis with simulation experiments and further analyze other components such as the Gating Mechanism, which we have found also play a role in rank collapse.

**Limitations and Future Work.** A limitation of this work is the exclusion of gating mechanisms from the theoretical analysis. Since gating mechanisms help to mitigate the effect of rank collapse, we expect their inclusion in the derivations would result in tighter upper bounds on the rank collapse measure for Mamba. Furthermore, one other line worth exploring is to consider the effect on lower bounds of MLPs in addition to skip connections and LayerNorm, Deriving a tighter lower bound that matches the results in our experiments is also a very promising avenue of research. Additionally, Remark 4.1 offers intriguing insights from a control-theoretic perspective, as it suggests that the $\lambda$ parameter can be interpreted as a control gain. We believe that further theoretical analysis of the stability and performance associated with $\lambda$-skip connections could lead to a more principled design of learning architectures. Finally, exploring alternative measures of rank collapse, like effective rank, and their correlation with our metric is a promising direction to expand on how our results generalize across metrics.

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

# A  APPENDIX

## A.1  PRELIMINARIES

We present a brief description of the sequence models considered in this work, namely attention (Stollenga et al. (2014); Luong et al. (2015); Song et al. (2016); Vaswani et al. (2023)) and recurrent blocks in State Space Models (Voelker et al. (2019); Gu et al. (2020; 2021; 2022b); Fu et al. (2023), and their corresponding architectural components (from Transformers Vaswani et al. (2023) and Mamba Gu and Dao (2024)).

## A.2  ATTENTION

Attention is considered to be the backbone of the Transformer architecture. This is thanks to its ability to capture correlations and dependencies between tokens in the sequence. In particular, the attention block can be expressed as follows:

$$O = \text{softmax}\left(\frac{XW_Q W_K^T X^T}{\sqrt{d_{QK}}}\right) XW_v \in \mathbb{R}^{N \times d}, \tag{8}$$

where the softmax operation is applied row-wise and $\sqrt{d_{QK}} \in \mathbb{R}$ acts as a sort of temperature parameter to modulate the strength of the interactions between tokens. $W_Q \in \mathbb{R}^{d \times d}$, $W_K \in \mathbb{R}^{d \times d}$ and $W_V \in \mathbb{R}^{d \times d}$ are the query, key and value matrices respectively, and $X \in R^{N \times d}$ is the input sequence, where $d$ is the embedding dimension and $N$ is the length of the sequence. In preparation for later sections, we also provide an alternative form for the attention block, as proposed by Dao and Gu (2024):

$$O = MV \tag{9}$$

with $V = XW_v \in \mathbb{R}^{N \times d}$ and $M = \text{softmax}(\frac{XW_Q W_K^T X^T}{\sqrt{d_{QK}}}) \in \mathbb{R}^{N \times N}$ for attention.

## A.3  RECURRENT BLOCK

SSMs lie their foundations in linear (often input-dependent) dynamical systems, which can be seen as linear RNNs (Orvieto et al. (2023)). In particular, the classic recurrent block for a generic SSM takes the following form:

$$\begin{cases} h_k = A_k h_{k-1} + B_k x_k \\ o_k = C_k h_k + D_k x_k \end{cases} \tag{10}$$

where $x_k \in R^d$ is the input at time $k$, where $k \in \{1, 2, ..., N\}$, $h_k \in R^H$ is the hidden state and $o_k \in R^d$ is the output of the SSM layer at time $k$. We note the connection with the Transformer's notation $x_k := X_{k,:}$, and similarly denote $O \in R^{N \times d}$ to be the output matrix of the SSM block, i.e. $o_k = O_{k,:}$. Here, given a matrix $A$, we indicate $A_{k,:}$ to be the k-th row.

One fundamental difference between attention and the recurrent block is that the attention mechanism does not have the capacity for memory, so the whole sequence of previous inputs $X$ has to be fed every time step. In contrast, the recurrent block in equation 10 is able to capture previous inputs' information, so only the last token has to be fed in. Yet, this expression also allows for a convolutional representation, where equation 10 is unrolled in time as:

$$o_k = \sum_{j=0}^{k} C_k \left(\prod_{l=0}^{k-j-1} A_{k-l}\right) B_j x_j + D_k x_k, \tag{11}$$

where $A_k \in \mathbb{R}^{H \times H}, B_k \in \mathbb{R}^{H \times d}, C_k \in \mathbb{R}^{d \times H}, D_k \in \mathbb{R}^{d \times d}$. The first versions of the SSM blocks consisted of Linear Time Invariant (LTI) representations, i.e., $A_k = A, B_k = B, C_k = C$ and $D_k = D$ for all time steps $k$. Recently, in order to improve the expressivity and the performance of such models, selective State Space Models have been proposed Gu and Dao (2024); Dao and Gu (2024). In these models, matrices $A_k, B_k$, and $C_k$ are input-dependent.

As shown in Dao and Gu (2024), the SSM update rule in equation 11 can be equivalently expressed as in equation 9, where $M$ is a lower triangular matrix with $M_{ji} = C_j \left( \prod_{l=0}^{j-i-1} A_{j-l} \right) B_i$ and $V = X$. In the reminder of the paper, this formulation will be used as it covers both the attention and the recurrent block.

According to Dao and Gu (2024), for selective SSMs where $A_t = \alpha_t I$ (e.g., Mamba-2), the matrix $M$ can be compactly written as $M^{(k)} = 1\text{SS}(\alpha) \odot \left( Y^{(k)} W_C W_B^\top Y^{(k)^T} \right)$, where $1\text{SS}(\alpha)$ is a lower-triangular 1-semiseparable matrix, which can be expressed as:

$$
1\text{SS}(\alpha) := \begin{bmatrix}
1 & & & & \\
\alpha_1 & 1 & & & \\
\alpha_2 \alpha_1 & \alpha_2 & 1 & & \\
\vdots & \vdots & \ddots & \ddots & \\
\alpha_{N-1} \dots \alpha_1 & \alpha_{N-1} \dots a_2 & \dots & \alpha_{N-1} & 1
\end{bmatrix}
\tag{12}
$$

## A.4  PROOF OF LOWER BOUND OF RANK COLLAPSE MEASURE

**Lemma A.1.** *Let $A \in R^{m \times n}$ and $B \in R^{n \times p}$ two matrices. Then it holds that $||AB||_F \geq \sigma_{min}(B)||A||_F$, where $\sigma_{min}(B)$ is the smallest singular value of $B$.*

*Proof.* From Fang et al. (1994), it holds that $||CD||_F \geq \sigma_{\min}(C)||D||_F$.
By the properties of the Frobenius norm, we have that $||AB||_F = ||B^\top A^\top||_F$. Hence, we have $||B^\top A^\top||_F \geq \sigma_{\min}(B^\top)||A^\top||_F$. The result then simply follows from the fact that $\sigma_{\min}(B^\top) = \sigma_{\min}(B)$. $\square$

**Theorem 4.1** (Lower Bound on Rank Collapse). *Let the input sequence $Y^{(0)}$ be such that $\mu(Y^{(0)})^2 \geq b$. If the skip connection strength $\lambda$ is chosen to satisfy*

$$
\lambda^2 - a(SC_M + |\lambda|)^2 > 0,
\tag{7}
$$

*then we can lower bound $\mu(Y^{(K)})$ by $\mu(Y^{(K)})^2 \geq a^K \mu(Y^{(0)})^2$ for all $K \in \mathbb{N}$.*

*Proof.* In the following, we denote $V_i^{(k)} := V_{:,i}^{(k)}$, $Y_i^{(k)} := Y_{:,i}^{(k)}$ and $\mu^{(k)} = \mu(Y^{(k)})$ for brevity.

$$
\tilde{\mu}^{(k+1)^2} = \left\| \tilde{Y}^{(k+1)} - \frac{\mathbf{1}\mathbf{1}^T}{N} \tilde{Y}^{(k+1)} \right\|_F^2 = \left\| M^{(k)}V^{(k)} + \lambda Y^{(k)} - \frac{\mathbf{1}\mathbf{1}^T}{N}(M^{(k)}V^{(k)} + \lambda Y^{(k)}) \right\|_F^2
$$

$$
= \lambda^2 \left\| Y^{(k)} - \frac{\mathbf{1}\mathbf{1}^T}{N} Y^{(k)} \right\|_F^2 + \sum_{i=1}^d \underbrace{\left\| M^{(k)}V_i^{(k)} - \mathbf{1}\mathbf{1}^T M^{(k)} \frac{V_i^{(k)}}{N} \right\|_2^2}_{\geq 0} +
$$

$$
+ \sum_{i=1}^d 2\lambda \left( M^{(k)}V_i^{(k)} - \mathbf{1}\mathbf{1}^T M^{(k)} \frac{V_i^{(k)}}{N} \right)^T \left( Y_i^{(k)} - \mathbf{1}\mathbf{1}^T \frac{Y_i^{(k)}}{N} \right)
$$

$$
\geq \lambda^2 \left\| Y^{(k)} - \frac{\mathbf{1}\mathbf{1}^T}{N} Y^{(k)} \right\|_F^2 + 2\lambda \sum_{i=1}^d \left[ \left( I - \frac{\mathbf{1}\mathbf{1}^T}{N} \right) M^{(k)}V_i^{(k)} \right]^T \left( I - \frac{\mathbf{1}\mathbf{1}^T}{N} \right) Y_i^{(k)}
$$

$$
= \lambda^2 \left\| Y^{(k)} - \frac{\mathbf{1}\mathbf{1}^T}{N} Y^{(k)} \right\|_F^2 + 2\lambda \sum_{i=1}^d V_i^{(k)^T} M^{(k)^T} \left( I - \frac{\mathbf{1}\mathbf{1}^T}{N} \right) Y_i^{(k)}.
$$

Continuing from here, we get:

$$
\tilde{\mu}^{(k+1)^2} \geq \lambda^2 \left\| Y^{(k)} - \frac{\mathbf{1}\mathbf{1}^T}{N} Y^{(k)} \right\|_F^2 - 2\lambda \sum_{i=1}^d ||M^{(k)}|| \left\| \left( I - \frac{1}{N}\mathbf{1}\mathbf{1}^T \right) \right\| \left\| V_i^{(k)^T} Y_i \right.
$$

$$= \lambda^2 \left\| Y^{(k)} - \frac{\mathbf{1}\mathbf{1}^T}{N} Y^{(k)} \right\|_F^2 - 2\lambda \|M^{(k)}\| \sum_{i=1}^d \sum_{j=1}^d C_{V_{:,i}}^T Y^{(k)^T} Y_i^{(k)}$$

$$= \lambda^2 \left\| Y^{(k)} - \frac{\mathbf{1}\mathbf{1}^T}{N} Y^{(k)} \right\|_F^2 - 2\lambda \|M^{(k)}\| \sum_{i=1}^d \sum_{j=1}^d C_{V_{ji}} Y_j^{(k)^T} Y_i^{(k)}$$

$$\geq \lambda^2 \left\| Y^{(k)} - \frac{\mathbf{1}\mathbf{1}^T}{N} Y^{(k)} \right\|_F^2 - 2\lambda \|M^{(k)}\| \max_{i,j} |C_{V_{ji}}| \sum_{i=1}^d \sum_{j=1}^d Y_j^{(k)^T} Y_i^{(k)}$$

$$= \lambda^2 \left\| Y^{(k)} - \frac{\mathbf{1}\mathbf{1}^T}{N} Y^{(k)} \right\|_F^2 - 2\lambda \|M^{(k)}\| \max_{i,j} |C_{V_{j,i}}| \sum_{i=1}^d \sum_{j=1}^d \sum_{l=1}^N Y_{lj}^{(k)} Y_{li}^{(k)}$$

$$= \lambda^2 \left\| Y^{(k)} - \frac{\mathbf{1}\mathbf{1}^T}{N} Y^{(k)} \right\|_F^2 - 2\lambda \|M^{(k)}\| \max_{i,j} |C_{V_{j,i}}| \sum_{l=1}^N \left( \sum_{j=1}^d Y_{lj}^{(k)} \right)^2$$

$$\geq \lambda^2 \left\| Y^{(k)} - \frac{\mathbf{1}\mathbf{1}^T}{N} Y^{(k)} \right\|_F^2 - 2\lambda \|M^{(k)}\| \max_{i,j} |C_{V_{j,i}}| \sum_{l=1}^N d \underbrace{\sum_{j=1}^d Y_{lj}^{(k)^2}}_{=1} \quad \text{(i)}$$

$$\geq \lambda^2 \left\| Y^{(k)} - \frac{\mathbf{1}\mathbf{1}^T}{N} Y^{(k)} \right\|_F^2 - 2\lambda \|M^{(k)}\| \max_{i,j} |C_{V_{j,i}}| Nd$$

$$\geq \lambda^2 \mu^{(k)^2} - 2\lambda \|M^{(k)}\|_F \, \|C_V\|_F Nd$$

where in (i) we used Cauchy-Schwartz inequality and the fact that we are using LayerNorm and hence every row of $Y^{(k)}$ has norm 1 and in the last step we used the definition of the rank collapse measure.

We now need to analyze the relationship between $\mu^{(k+1)}$ and $\tilde{\mu}^{(k+1)}$.

$$\mu^{(k+1)^2} = \left\| D^{(k+1)} \tilde{Y}^{(k+1)} - \frac{\mathbf{1}\mathbf{1}^T}{N} D^{(k+1)} \tilde{Y}^{(k+1)} \right\|_F^2 = \left\| \left( I - \frac{\mathbf{1}\mathbf{1}^T}{N} \right) D^{(k+1)} \tilde{Y}^{(k+1)} \right\|_F^2$$

$$= \sum_{j=1}^N \sum_{i=1}^d \left( \sum_{l=1}^N \left( I - \frac{\mathbf{1}\mathbf{1}^T}{N} \right)_{jl} \frac{\tilde{Y}_{il}^{(k+1)}}{\|\tilde{Y}_{i,:}^{(k+1)}\|_2} \right)^2$$

$$\geq \sum_{j=1}^N \sum_{i=1}^d \left( \sum_{l=1}^N \left( I - \frac{\mathbf{1}\mathbf{1}^T}{N} \right)_{jl} \frac{\tilde{Y}_{il}^{(k+1)}}{\max_{i \in [N]} \|\tilde{Y}_{i,:}^{(k+1)}\|_2} \right)^2$$

$$\geq \frac{1}{\max_{i \in [N]} \|\tilde{Y}_{i,:}^{(k+1)}\|_2^2} \sum_{j=1}^N \sum_{i=1}^d \left( \sum_{l=1}^N \left( I - \frac{\mathbf{1}\mathbf{1}^T}{N} \right)_{jl} \tilde{Y}_{il}^{(k+1)} \right)^2$$

$$= \frac{1}{\max_{i \in [N]} \|\tilde{Y}_{i,:}^{(k+1)}\|_2^2} \tilde{\mu}^{(k+1)^2}$$

Combining the previous two results and recalling that $C_M = \sup_k \|M^{(k)}\|_F$ and $S = \sup_k \|C_V\|_F$, we then get:

$$\mu^{(k+1)^2} \geq \frac{1}{\max_{i \in [N]} \|\tilde{Y}_{i,:}^{(k+1)}\|_2^2} \left( \lambda^2 \mu^{(k)^2} - 2\lambda N d S C_M \right)$$

We will now proceed in upper bounding $\max_{i\in[N]} ||\tilde{Y}_{i,:}^{(k+1)}||_2^2$.

$$
\begin{aligned}
\max_{i\in[N]} ||\tilde{Y}_{i,:}^{(k+1)}||_2^2 &= \max_{i\in[N]} \sum_j \tilde{Y}_{ij}^{(k+1)^2} = \max_{i\in[N]} \sum_j \left( \sum_l \left( M_{il}^{(k)} V_{lj}^{(k)} + \lambda Y_{lj}^{(k)} \right) \right)^2 \\
&\leq \max_{i\in[N]} \sum_j \sum_l \left( M_{il}^{(k)^2} V^{(k)^2} + \lambda_{lj}^2 Y_{lj}^{(k)^2} + 2\lambda M_{il}^{(k)} V_{lj}^{(k)} Y_{lj}^{(k)} \right) \\
&= \max_{i\in[N]} \left( \sum_l M_{il}^{(k)^2} \underbrace{\sum_j V_{lj}^{(k)^2}}_{\leq S^2} + \lambda^2 + 2\lambda \sum_l M_{il}^{(k)} \sum_j V_{lj}^{(k)} Y_{lj}^{(k)} \right) \\
&\leq \max_{i\in[N]} \left( S^2 \underbrace{\sum_l M_{il}^{(k)^2}}_{\leq C_M^2} + \lambda^2 + 2\lambda \sum_l M_{il}^{(k)} \sum_m C_{V_{lm}}^{(k)} \underbrace{\sum_j Y_{mj}^{(k)} Y_{lj}^{(k)}}_{\leq 1} \right) \\
&\leq \max_{i\in[N]} \left( S^2 C_M^2 + \lambda^2 + 2\lambda C_M S \right) \\
&\leq (C_M S + \lambda)^2
\end{aligned}
$$

By putting everything together, we get:

$$
\mu^{(k+1)^2} \geq \frac{1}{(SC_M + |\lambda|)^2} \left( \lambda^2 \mu^{(k)^2} - 2\lambda N d S C_M \right)
$$

To guarantee that $\mu^{(k+1)^2} \geq a\mu^{(k)^2}$ for some $a < 1$, we can simply lower bound the right hand side. Hence, we want that $\frac{1}{(SC_M+|\lambda|)^2} \left( \lambda^2 \mu^{(k)^2} - 2\lambda N d S C_M \right) \geq a\mu^{(k)^2}$.
Hence, we need:

$$
\left( \lambda^2 - a(SC_M + |\lambda|)^2 \right) \mu^{(k)^2} \geq 2\lambda N d S C_M
$$

To do so, we need to guarantee that $\lambda^2 - a(SC_M + |\lambda|)^2 > 0$. Hence, if $\lambda$ satisfies the condition above and we have that $\mu^{(k)^2} \geq \frac{2\lambda N d S C_M}{\lambda^2 - a(SC_M+|\lambda|)^2}$, then $\mu^{(k+1)^2} \geq a\mu^{(k)^2}$. Note that in order to satisfy the condition on $\mu^{(k)^2}$ up until time $K$, we need to choose $\mu^{(0)^2} \geq \frac{1}{a^K} \frac{2\lambda N d S C_M}{\lambda^2 - a(SC_M+|\lambda|)^2}$, which concludes the proof.

$\square$

In the following, we provide a brief discussion on the key variables of interest in the bound above, their typical values in practice, their relationship and implications on the bound, in particular focusing describing how it is possible to choose suitable values for $\lambda$. In the presented bound, the key variables of interest are $\lambda$, $a$ and $\mu(Y^{(k)})$. Specifically, we aim for $\mu(Y^{(k)})$ to be as far from 0 as possible, as values close to 0 indicate rank collapse. Furthermore, as we mentioned in the main text, the ideal value of $a$ would be 1 (since this would guarantee that the rank collapse metric is non decreasing over layers), although in order to satisfy Equation 7 this value cannot be chosen. In practice, the typical value of $\lambda$ is 1. The key relationship between $a$ and $\lambda$ is the following: in order to guarantee values of $a$ closer and closer to 1 (and hence to ensure higher values of the rank collapse metric at the final layer) we must choose larger values for $|\lambda|$. Additionally, $N$ represents the input sequence length, which varies based on the task. For example, $N$ might be on the order of tens for simple question answering tasks, but it could scale to hundreds or thousands when summarizing a long document. $d$ instead represents the embedding dimension, typically in the order of

tens or hundreds. Regarding the selection of $\lambda$, we propose two possible approaches: treating it as a hyperparameter or making it learnable. In the first approach, $\lambda$ would be chosen through a standard hyperparameter optimization procedure, testing different values and evaluating their impact on both performance and the rank collapse measure. While effective, this method requires multiple training runs to identify the best value. In contrast, the second approach—making $\lambda$ learnable—offers significant advantages. It automates the process of finding an optimal $\lambda$, eliminating the need for manual hyperparameter tuning and requiring only a single training run. This is both more efficient and practical. For these reasons, we adopted the learnable $\lambda$ approach in our experiments, as illustrated in Table 1.

### A.5 Theorem on Rank Collapse in Self-Attention Only Transformers with LayerNorm

We first define some notation that is used in the Theorem. First, we denote $\phi^{(t)} = Y_{i,:}^{(t)}, Y_{j,:}^{(t)}\rangle$ where $\langle, \cdot, \cdot, \rangle$ is the inner product. Then, $\mathcal{G}$ represents the graph induced by the attention mask. In particular, if there is a directed edge from $j$ to $i$, this means that token $i$ attends to token $j$. A Quasi-Strongly Connected graph is a graph $\mathcal{G}$ in which there exists a node from which every other node in the directed graph $\mathcal{G}$ is reachable (i.e. there exist a directed path connecting the two nodes). Finally, the authors consider the following assumptions:

**Assumption A.1.** *$\mathcal{G}$ contains self-loops, i.e. every token in the sequence attends to itself*

**Assumption A.2.** *There exists a constant $C \in \mathbb{R}$ such that $\max_{t\in\mathbb{N}} \left\{ ||W_Q^{(t)}||_2, ||W_K^{(t)}||_2 \right\} \leq C$*

**Theorem A.2** (Corollary 1, Wu et al. (2024a)). *Consider the architecture defined in 6 by choosing $\lambda = 0$, i.e. a Self-Attention Network with LayerNorm and no skip connection. Let $\mathcal{G}$ be a quasi-strongly connected graph. Under A.1 and A.2, if $W_V^{(t)}$ is orthogonal for all $t \geq 0$ and $\phi^{(0)} \geq 0$, there exist $C > 0$ and $\epsilon > 0$ such that $N\epsilon < 1$ and*

$$\mu(Y^{(K)}) \leq C(1 - \epsilon^{2r})^{\frac{K}{2r}}, \quad \forall K \geq 0.$$

*where $r$ is the diameter of $\mathcal{G}$, meaning that tokens converge to a common point on $\mathbb{S}^{d-1}$ exponentially.*

### A.6 Theorem on Rank Collapse in Self-Attention only Transformers without LayerNorm

Let $\text{res}(Y) = Y - \frac{\mathbf{1}\mathbf{1}^\top Y}{N}$. Note that with this notation, we can redefine $\mu(Y) = ||\text{res}(Y)||_F$. Then, it holds that:

**Theorem A.3** (Rank Collapse Transformers Dong et al. (2023)). *For any single-head Self-Attention network consisting of $K$ layers with $\left\|\mathbf{W}_Q^{(k)} W_K^{(k)^\top}\right\|_1 \left\|\mathbf{W}_V^{(k)}\right\|_{1,\infty} \leq \beta$ and for a term $\gamma$ that depends on the attention entries, we have that*

$$\| \text{res}(Y^{(K)})\|_{1,\infty} \leq \left( \frac{4\gamma\beta}{\sqrt{d_{qk}}} \right)^{\frac{3^K-1}{2}} \| \text{res}(Y^{(0)})\|_{1,\infty}^{3^K}$$

*which amounts to a doubly exponential convergence to a rank-1 matrix.*

**Corollary A.3.1.** *Under the same conditions of Theorem A.3, it holds that*

$$|| \text{res}(Y^{(K)})||_F \leq \left( \frac{4\gamma\sqrt{\min(N,d)}\beta}{\sqrt{d_{qk}}} \right)^{\frac{3^K-1}{2}} || \text{res}(Y^{(0)})||_F^3.$$

*Proof.* Follows straightforward from the relations between Frobenius norm and the 1- and $\infty$-norms, i.e. given a matrix we have $A \in R^{N \times d}$ $||A||_F \geq ||A||_1, ||A||_F \geq ||A||_\infty, ||A||_F \leq \min(N,d)||A||_1, ||A||_F \leq \min(N,d)||A||_\infty$. $\qquad\square$

### A.7 PROOF OF RANK COLLAPSE FOR SELECTIVE SSMS WITH LAYERNORM BUT WITHOUT SKIP CONNECTIONS

**Lemma A.4.** *If $\alpha \leq 1$, We have that $M_{ij}^{(k)} \geq \lambda_{min} \cdot \phi^{(k)} \alpha^N$ for $i \geq j$*

*Proof.* If $i \geq j$, we have that:

$$M_{ij}^{(k)} = \alpha^{i-j} Y_{i,:}^{(k)^\top} W_C W_B^\top Y_{j,:}^{(k)} \geq \alpha^{i-j} \lambda_{min} \langle Y_{i,:}^{(k)}, Y_{j,:}^{(k)} \rangle \geq \alpha^{j-i} \lambda_{min} \cdot \phi^{(k)}$$

Note that the worst case is when $i - j = N - 1$, which leads to the desired lower bound by lower bounding $\alpha^{i-j} \geq \alpha^{N-1} \geq \alpha^N$ since $a \leq 1$. $\square$

**Lemma A.5.** *We have that $D_{i,i}^{(k)} \geq \frac{1}{N^{\frac{3}{2}} \lambda_{max}} \forall k$*

*Proof.* From the definition of LayerNorm, we have $D_{i,i}^{(k)} = \frac{1}{||\tilde{Y}_{i,:}^{(k)}||_2}$. Furthermore, we have that

$$||\tilde{Y}_{i,:}^{(k)}||_2 = ||\sum_{j=1}^N M_{ij}^{(k-1)} Y_{j,:}^{(k-1)}||_2 \leq \sum_{j=1}^N M_{ij}^{(k-1)} \underbrace{||Y_{j,:}^{(k-1)}||_2}_{=1} \leq N\lambda_{max}. \qquad \square$$

**Theorem 4.3** (Rank Collapse for selective SSMs without skip connection). *Let $\phi^{(k)} = \min_{i,j \in [N]} \langle Y_{i,:}^{(k)}, Y_{j,:}^{(k)} \rangle$, where $\langle \cdot, \cdot \rangle$ indicates the inner product. Under Assumption 4.1 and if $c \leq \phi^{(0)} < 1$ for some $c > 0 \in \mathbb{R}$, $\lambda_{min} > 0$, $\sum_j M_{ij} \geq 1 \ \forall i$, then it holds that:*

$$\mu(Y^{(K)}) \leq \sqrt{N} \left(1 - c^2 \lambda_{min}^2 \alpha^{2N}\right)^K \ \forall K \geq 0,$$

*where $\lambda_{min}$ and $\lambda_{max}$ the minimum and maximum eigenvalues of $\frac{W_B W_C^\top + W_C W_B^\top}{2}$.*

*Proof.* For the fist part of the proof, we will follow the proof of Corollary 1 in Wu et al. (2024a):

$$\begin{aligned}
\phi^{(k+1)} &= \min_{i,j \in [N]} \langle Y_{i,:}^{(k+1)}, Y_{j,:}^{(k+1)} \rangle = \langle Y_{i^{(k+1)},:}^{(k+1)}, Y_{j^{(k+1)},:}^{(k+1)} \rangle \\
&= \left\langle \sum_{k_1=1}^N \left( D^{(k)} M^{(k)} \right)_{i^{(k+1)} k_1} Y_{k_1,:}^{(k)}, \sum_{l_1=1}^N \left( D^{(k)} M^{(k)} \right)_{j^{(k+1)} l_1} Y_{l_1,:}^{(k)} \right\rangle \\
&\geq \frac{1}{N\lambda_{max}} \left\langle \sum_{k_1=1}^N M_{i^{(k+1)} k_1}^{(k)} Y_{k_1,:}^{(k)}, \sum_{l_1=1}^N M_{j^{(k+1)} l_1}^{(k)} Y_{l_1,:}^{(k)} \right\rangle
\end{aligned}$$

where the last inequality follows from Lemma A.5.

From here, using Lemma A.4 and the assumption that $\lambda_{max} \leq \frac{1}{N}$ we have that:

$$\begin{aligned}
\phi^{(k+1)} &\geq \sum_{k_1=1}^N \sum_{l_1=1}^N M_{i^{(k+1)} k_1}^{(k)} M_{j^{(k+1)} l_1}^{(k)} \langle Y_{k_1,:}^{(k)}, Y_{l_1,:}^{(k)} \rangle \\
&= \sum_{m=1}^N M_{i^{(k+1)} m}^{(k)} M_{j^{(k+1)} m}^{(k)} \underbrace{\langle Y_{m,:}^{(k)}, Y_{m,:}^{(k)} \rangle}_{=1} + \sum_{k_1=1}^N \sum_{l_1=1, l_1 \neq k_1}^N M_{i^{(k+1)} k_1}^{(k)} M_{j^{(k+1)} l_1}^{(k)} \underbrace{\langle Y_{k_1,:}^{(k)}, Y_{l_1,:}^{(k)} \rangle}_{\geq \phi^{(k)}} \\
&\geq \lambda_{min}^2 \phi^{(k)^2} \alpha^{2N} + (1 - \lambda_{min}^2 \alpha^{2N} \phi^{(k)^2}) \phi^{(k)} \\
&\geq c^2 \lambda_{min}^2 \alpha^{2N} + \left(1 - c^2 \lambda_{min}^2 \alpha^{2N}\right) \phi^{(k)}
\end{aligned}$$

where the last two inequalities follow since $\sum_j M_{ij} \geq 1 \ \forall i$ and $\phi^{(k)} \geq c$ respectively. Note also that by iterating the above recurrence starting from $t = 1$, it is easy to show by induction that since $\phi^{(0)} \geq c$, then $\phi^{(k)} \geq c \ \forall k$. By rearranging the terms, we have that:

$$1 - \phi^{(k+1)} \leq \left(1 - c^2 \lambda_{\min}^2 \alpha^{2N}\right)\left(1 - \phi^{(k)}\right)$$

By recursively unrolling the relationship, we get:

$$1 - \phi^{(K)} \leq \left(1 - c^2 \lambda_{\min}^2 \alpha^{2N}\right)^K \left(1 - \phi^{(0)}\right)$$
$$\leq \left(1 - c^2 \lambda_{\min}^2 \alpha^{2N}\right)^K$$

since $0 < c \leq \phi^{(0)} < 1$

Since by definition, $1 - \phi^{(k)} \geq 1 - \langle Y_{i,:}^{(k)}, Y_{j,:}^{(k)} \rangle \geq \|Y_{i,:}^{(k)} - Y_{j,:}^{(k)}\|_2^2 / 2$ it follows that:

$$\mu(Y^{(K)}) = \|Y^{(K)} - \mathbf{1}\mathbf{1}^\top Y^{(K)}/N\|_F = \sqrt{\sum_{i=1}^N \|Y_{i,:}^{(K)} - \mathbf{1}^\top Y^{(K)}/N\|_2^2}$$
$$= \sqrt{\frac{1}{2N} \sum_{i=1}^N \sum_{j=1}^N \|Y_{i,:}^{(K)} - Y_{j,:}^{(K)}\|_2^2}$$
$$\leq \sqrt{N}\left(1 - c^2 \lambda_{\min}^2 \alpha^{2N}\right)^K$$

$\square$

## A.8  RANK COLLAPSE FOR LTI SSMs WITH NO SKIP CONNECTIONS AND LAYERNORM

**Lemma A.6.**
$$||Y^{(k)}||_F \leq ||W||^k ||Y^0||_F$$

*Proof.* We have that $||Y^{(k+1)}||_F = ||M^{(k)}Y^{(k)}||_F \leq ||W|| ||Y^{(k)}||_F$. By simply solving the recurrence we get the desired result. $\square$

We now analyze rank collapse for Structured LTI SSMs. For simplicity of exposition, here we neglect the gating mechanism and focus on the recurrent block. In particular, we will analyze a network consisting of $K$ layers of stacked Structured LTI SSM blocks. We recall that for structured LTI SSM blocks, which satisfy with $A_t = a_t I$, we have that the matrix $M$ takes the form $M^{(k)} =$

$W_A^{(k)} \odot (W_C^{(k)} W_B^{(k)^T}) := W^{(k)}$, where $W_A = \begin{bmatrix} 1 & & & \\ a_1 & 1 & & \\ a_2 a_1 & a_2 & 1 & \\ \vdots & \vdots & \ddots & \ddots \\ a_{N-1}\dots a_1 & a_{N-1}\dots a_2 & \dots & a_{N-1} & 1 \end{bmatrix}$.

We first consider the case where all the layers have the same parameters, i.e. it holds that $W^{(k)} = W \ \forall k$ for some matrix $W$. We then have the following result:

**Theorem A.7** (Rank Collapse Structured LTI SSMs). *For a $K$ layer network of only Structured LTI SSM blocks where all layers have the same parametrization, it holds that $\mu(Y^{(k)}) \leq ||W||^K ||Y^{(0)}||_F$*

*Proof.* We have that:

$$M^{(k)}Y^{(k)} - \mathbf{1}\gamma_{k+1} = \left(I - \frac{\mathbf{1}\mathbf{1}^\top}{N}\right)WY^{(k)}$$

From here, by using Lemma A.6, we get:

$$\mu(Y^{(k+1)}) = ||M^{(k)}Y^{(k)} - 1\gamma_{k+1}||_F$$
$$\leq \left|\left|I - \frac{11^\top}{N}\right|\right| \left|\left|WY^{(k)}\right|\right|_F \leq ||Y^{(k+1)}||_F$$
$$\leq ||W||^{k+1}||Y^{(0)}||_F$$

$\square$

Hence, whenever $||W||_F < 1$, we have rank collapse for the case of structured LTI-SSMs. Note that however here the overall dynamics and behavior is different: the convergence is exponential rather than doubly exponential like in the Attention case. This is due to the fact that the matrix $M$ is not input dependent, which is what causes the double exponential convergence. We will see that for selective SSMs, which also have a similar input dependence to attention (apart from the softmax) for the matrix $M$, the convergence will again be doubly exponential. Furthermore, note that here the initial input sequence does not have any influence on rank collapse, whereas in the attention case this was not the case (since there was also a doubly exponential dependence on $||\text{res}(Y^{(0)})||_F$). Again, we will see a similar behavior in the selective SSM setting.

For the case where we have different parametrizations for different layers, we then have:

**Theorem A.8** (Rank Collapse Structured LTI SSMs General). *For a $K$ layer network of only Structured LTI SSM blocks it holds that $\mu(Y^{(K)}) \leq \prod_{k=1}^{K} ||W^{(k)}|| \, ||Y^{(0)}||_F$*

*Proof.* From the proof of Lemma A.6, we have that $||Y^{(k+1)}||_F = ||M^{(k)}Y^{(k)}||_F \leq ||W^{(k)}|| \, ||Y^{(k)}||_F$. Applying this recursively gives us $||Y^{(K)}||_F \leq \prod_{k=1}^{K} ||W^{(k)}|| \, ||Y^{(0)}||_F$. Then, similarly to the proof of TheoremA.7, we have:

$$\mu(Y^{(K+1)}) = ||M^{(K)}Y^{(K)} - 1\gamma_{K+1}||_F$$
$$\leq \left|\left|I - \frac{11^\top}{N}\right|\right| \left|\left|W^{(K)}Y^{(K)}\right|\right|_F \leq \sqrt{N}||Y^{(K+1)}||_F$$
$$\leq \prod_{k=1}^{K+1} ||W^{(k)}|| \, ||Y^{(0)}||_F$$

$\square$

In this case, one simple sufficient condition to have exponential convergence of the rank collapse measure is that $||W^{(k)}||_F < 1 \, \forall k$.

## A.9 PROOF OF UPPER BOUND OF RANK COLLAPSE MEASURE FOR SELECTIVE SSMs WITHOUT LAYERNORM AND SKIP CONNECTIONS

**Lemma A.9.** *For a selective SSM, it holds that:*

$$||Y^{(k)}||_F \leq \left(\sqrt{N}||W_{BC}||_F\right)^{\frac{3^k-1}{2}} ||Y^{(0)}||_F^{3^k}$$

*where $W_{BC} = W_C W_B^\top$*

*Proof.* We have that:

$$||Y^{(k+1)}||_F = ||M^{(k)}Y^{(k)}||_F \leq ||M^{(k)}||_F \, ||Y^{(k)}||_F$$
$$= ||1\text{SS}(\alpha) \odot \left(Y^{(k)}W_C W_B^\top Y^{(k)^T}\right)||_F \, ||Y^{(k)}||_F$$
$$\leq ||1\text{SS}(\alpha)||_F ||Y^{(k)}W_C W_B^\top Y^{(k)^T}||_F \, ||Y^{(k)}||_F$$
$$\leq \sqrt{N}||W_{BC}||_F ||Y^{(k)}||_F^3$$

By simply solving the recurrence we get the desired result. $\square$

**Theorem A.10** (Rank Collapse for Selective SSMs). *Given a selective SSM with same parametrization for each layer with no skip connection, it holds that:*

$$\mu(Y^{(k)}) \leq \left(\sqrt{N}||W_{BC}||_F\right)^{\frac{3^{k-1}+1}{2}} ||Y^{(0)}||_F^{3^k}.$$

*Proof.* We start by observing that:

$$\mu(Y^{(k)}) = \left|\left|\left(I - \frac{11^\top}{N}\right) M^{(k-1)}Y^{(k-1)}\right|\right|_F \leq \sqrt{N}||M^{(k-1)}||_F \, ||Y^{(k-1)}||_F$$
$$\leq \sqrt{N}||W_{BC}|| \, ||Y^{(k-1)}||_F^3$$

By using Lemma A.9, we get that:

$$\mu(Y^{(k)}) \leq \left(\sqrt{N}||W_{BC}||_F\right)^{\frac{3^{k-1}+1}{2}} ||Y^{(0)}||_F^{3^k}$$

proving the result. □

### A.10 PROOF OF COUNTEREXAMPLE FOR LTI SSMs

**Structured LTI SSMs.** Consider the system $A_t = A = \alpha I, B_t = B = I$ and $C_t = C = I$, and let $N = 2$ and $d = 2$. By using the relation $M_{ji} = C_j^\top A_j \ldots A_{i+1} B_i$ whenever $j \geq i$ (Dao and Gu, 2024), we get that $M + \lambda I = \begin{pmatrix} 1 + \lambda & 0 \\ \alpha & 1 + \lambda \end{pmatrix}$. Let us denote this system, equipped with LayerNorm, as [Sys-1].

**Proposition A.10.1.** *Given [Sys-1], if we choose $\alpha = 2$, i.e. $M = \begin{pmatrix} 1 & 0 \\ 2 & 1 \end{pmatrix}$, $\lambda > -2$ and $Y^{(0)} = I$, then we have that $\mu(Y^{(k)}) \to 0$. Conversely, given the choices above with $\lambda < -2$ (with $\lambda \neq -1$, then rank collapse does not happen, i.e. $\mu(Y^{(k)}) \not\to 0$.*

*Proof.* We will prove the theorem by induction.
In particular, we will prove that $Y^{(k)}$ will take the following form:

$$Y^{(k)} = \begin{pmatrix} 1 & 0 \\ \sqrt{\frac{\alpha_k - 1}{\alpha_k}} & \frac{1}{\sqrt{\alpha_k}} \end{pmatrix} \tag{13}$$

where $\alpha_0 = 1$ and $\alpha_k$ satisfy the recurrence: $\alpha_{k+1} = \alpha_k \left(1 + \frac{4}{(1+\lambda)^2}\right) + \frac{4}{1+\lambda}\sqrt{\alpha_k - 1}\sqrt{\alpha_k}$. For $\lambda > -1$, we simply have that $\alpha_{k+1} \geq \alpha_k \left(1 + \frac{4}{(1+\lambda)^2}\right)$, implying that $\alpha_k \geq \alpha_0 \left(1 + \frac{4}{(1+\lambda)^2}\right)^k$, which causes $\alpha_k$ to diverge. Similarly, if $-2 < \lambda < -1$, we have that $\alpha_{k+1} \geq \alpha_k \left(1 + \frac{2}{1+\lambda}\right)^2$, i.e. $\alpha_k \geq \alpha_0 \left(\frac{3+\lambda}{1+\lambda}\right)^{2k}$, which also diverges. This implies that $Y^{(k)} \to \begin{pmatrix} 1 & 0 \\ 1 & 0 \end{pmatrix}$, which is a rank one matrix, hence also implying that $\mu(Y^{(k)}) \to 0$.

Let's consider now the case where $\lambda < -2$. We will prove that the sequence $\alpha_k$ remains bounded from above. Note that this automatically proves that rank collapse does not happen, since a necessary condition for this is that $\alpha_k \to \infty$. Let define $r = \frac{1}{1+\lambda}$ and $\beta = \frac{1}{1-r^2} > 1$ for $\lambda < -2$. We prove by induction that $\alpha_k \leq \beta \forall k$. Since $\beta > 1$, the statement clearly holds for $k = 0$. Suppose now that the statement holds for $k$. We prove that the statement holds for $k + 1$. We have that $\alpha_{k+1} \leq \beta(1 + 4r^2) + 4r\sqrt{\beta}\sqrt{\beta - 1}$. We want the quantity on the right hand side to be smaller than

$\beta$. This is equivalent to saying that $r^2\beta \leq -r\sqrt{\beta}\sqrt{\beta-1}$. By squaring both sides and solving the inequality we can see that $\beta = \frac{1}{1-r^2}$ satisfies this inequality.

We now go on to prove the above statement in Equation 13 about the structure of $Y^{(k)}$ using induction. Clearly, the statement is true for $k = 0$. Now, let's assume the statement holds for $k$. We will prove that then the statement holds for $k+1$. We have that:

$$\tilde{Y}^{(k+1)} = (M+\lambda I)Y^{(k)} = \begin{pmatrix} 1+\lambda & 0 \\ 2 & 1+\lambda \end{pmatrix}\begin{pmatrix} 1 & 0 \\ \sqrt{\frac{\alpha_k-1}{\alpha_k}} & \frac{1}{\sqrt{\alpha_k}} \end{pmatrix} = \begin{pmatrix} 1+\lambda & 0 \\ \frac{(1+\lambda)\sqrt{\alpha_k-1}+2\sqrt{\alpha_k}}{\sqrt{\alpha_k}} & \frac{1+\lambda}{\sqrt{\alpha_k}} \end{pmatrix}$$

Hence, after we apply LayerNorm to $\tilde{Y}^{(k+1)}$, by letting $\gamma = \left(4+(1+\lambda)^2\right)\alpha_k + 4(1+\lambda)\sqrt{\alpha_k}\sqrt{\alpha_k-1}$, we get that:

$$Y^{(k+1)} = \begin{pmatrix} 1 & 0 \\ \frac{(1+\lambda)\sqrt{\alpha_k-1}+2\sqrt{\alpha_k}}{\sqrt{\gamma}} & \frac{1+\lambda}{\sqrt{\gamma}} \end{pmatrix}$$

Choosing $\alpha_{k+1} = \left(1+\frac{4}{(1+\lambda)^2}\right)\alpha_k + \frac{4}{1+\lambda}\sqrt{\alpha_k-1}\sqrt{\alpha_k}$ clearly results to $k+1$ being of the form

$$Y^{(k+1)} = \begin{pmatrix} 1 & 0 \\ \sqrt{\frac{\alpha_{k+1}-1}{\alpha_{k+1}}} & \frac{1}{\sqrt{\alpha_{k+1}}} \end{pmatrix}$$

which proves the induction hypothesis. $\qquad\square$

### A.11 PROOF OF COUNTEREXAMPLE FOR SELECTIVE SSMs

**Proposition 4.3.1.** *Given [Sys-2], if we then choose $\lambda > -\frac{3}{2}$, $M$ as above and $Y^{(0)} = \begin{pmatrix} 1 & 0 \\ \frac{\alpha_0}{\sqrt{\alpha_0^2+\beta_0^2}} & \frac{\beta_0}{\sqrt{\alpha_0^2+\beta_0^2}} \end{pmatrix}$ for any $\alpha_0, \beta_0$ with $\alpha_0 > 0$, then we have that $\mu(Y^{(k)}) \to 0$. On the other hand, if we choose $\lambda < -\frac{3}{2}$ and $M$ and $Y^{(0)}$ as above, then rank collapse is avoided.*

*Proof.* Again, we proceed by induction. In particular, we will show that under the assumptions of the theorem, $Y^{(k)}$ takes the following form:

$$Y^{(k)} = \begin{pmatrix} 1 & 0 \\ \frac{(2+\lambda)^k\alpha_0}{\sqrt{(2+\lambda)^{2k}\alpha_0^2+(1+\lambda)^{2k}\beta_0^2}} & \frac{(1+\lambda)^k\beta_0}{\sqrt{(2+\lambda)^{2k}\alpha_0^2+(1+\lambda)^{2k}\beta_0^2}} \end{pmatrix}$$

Note that if this result holds, as long as $\alpha_0 > 0$ and $|2+\lambda| > |1+\lambda|$ (which happens iff $\lambda > -\frac{3}{2}$), we have that $Y^{(k)} \to \begin{pmatrix} 1 & 0 \\ 1 & 0 \end{pmatrix}$, which is a rank one matrix, hence also implying that $\mu(Y^{(k)}) \to 0$. On the other hand, if $\lambda < -\frac{3}{2}$ we have that $Y^{(k)}$, as k increases, tends to oscillate between the matrices $\begin{pmatrix} 1 & 0 \\ 0 & 1 \end{pmatrix}$ and $\begin{pmatrix} 1 & 0 \\ 0 & -1 \end{pmatrix}$. Although in this case the limit of $Y^{(k)}$ does not exist, we have that both the matrices above have full rank and hence rank collapse does not happen.

We now proceed to prove the statement above. Clearly, the statement is true for $k = 0$. Let's suppose that the statement holds for $k$. We will prove that it holds for $k+1$. We have that under the choice of $M$ and $\lambda$ for the theorem, we have that:

$$\tilde{Y}^{(k+1)} = \begin{pmatrix} 1+\lambda & 0 \\ \frac{(2+\lambda)^k\alpha_0}{\sqrt{(2+\lambda)^{2k}\alpha_0^2+(1+\lambda)^{2k}\beta_0^2}} & 1+\lambda \end{pmatrix}\begin{pmatrix} 1 & 0 \\ \frac{(2+\lambda)^k\alpha_0}{\sqrt{(2+\lambda)^{2k}\alpha_0^2+(1+\lambda)^{2k}\beta_0^2}} & \frac{(1+\lambda)^k\beta_0}{\sqrt{(2+\lambda)^{2k}\alpha_0^2+(1+\lambda)^{2k}\beta_0^2}} \end{pmatrix}$$

$$= \begin{pmatrix} \frac{1+\lambda}{\sqrt{(2+\lambda)^{2k}\alpha_0^2+(1+\lambda)^{2k}\beta_0^2}} & \frac{0}{\sqrt{(2+\lambda)^{2k}\alpha_0^2+(1+\lambda)^{2k}\beta_0^2}} \\ \frac{(2+\lambda)^{k+1}\alpha_0}{\sqrt{(2+\lambda)^{2k}\alpha_0^2+(1+\lambda)^{2k}\beta_0^2}} & \frac{(1+\lambda)^{k+1}\beta_0}{\sqrt{(2+\lambda)^{2k}\alpha_0^2+(1+\lambda)^{2k}\beta_0^2}} \end{pmatrix}$$

Hence, after applying LayerNorm to $\tilde{Y}^{(k+1)}$, we get that:

$$Y^{(k)} = \begin{pmatrix} \frac{1}{\sqrt{(2+\lambda)^{2k+2}\alpha_0^2+(1+\lambda)^{2k+2}\beta_0^2}} & \frac{0}{\sqrt{(2+\lambda)^{2k+2}\alpha_0^2+(1+\lambda)^{2k+2}\beta_0^2}} \\ \frac{(2+\lambda)^{k+1}\alpha_0}{\sqrt{(2+\lambda)^{2k+2}\alpha_0^2+(1+\lambda)^{2k+2}\beta_0^2}} & \frac{(1+\lambda)^{k+1}\beta_0}{\sqrt{(2+\lambda)^{2k+2}\alpha_0^2+(1+\lambda)^{2k+2}\beta_0^2}} \end{pmatrix}$$

which proves the claim and finishes the proof. $\qquad\square$

## A.12 PROOF OF THE TIGHTNESS OF THE LOWER BOUND IN THEOREM 4.1

**Proposition 4.3.2.** *There exist an architecture of the form of Equation 6 such that, when choosing $\lambda$ to satisfy Equation 7, i.e. $|\lambda| = \Omega(\frac{a}{1-a})$, it holds that $\mu(Y^{(k)})^2 = O\left(a^k\mu(Y^{(0)})^2\right)$.*

*Proof.* We again consider [Sys-2] from Proposition 4.3.1 with $\alpha_0 = \beta_0 = \frac{1}{2}$. First, we calculate the value of the rank collapse metric at every layer. Following the proof of Proposition 4.3.1, where we saw that $Y^{(k)} = \begin{pmatrix} \frac{1}{\sqrt{(2+\lambda)^{2k}\alpha_0^2+(1+\lambda)^{2k}\beta_0^2}} & 0 \\ \frac{(2+\lambda)^k\alpha_0}{\sqrt{(2+\lambda)^{2k}\alpha_0^2+(1+\lambda)^{2k}\beta_0^2}} & \frac{(1+\lambda)^k\beta_0}{\sqrt{(2+\lambda)^{2k}\alpha_0^2+(1+\lambda)^{2k}\beta_0^2}} \end{pmatrix}$, a simple calculation leads to $\mu(Y^{(k)}) = \frac{1}{1+(\frac{2+\lambda}{1+\lambda})^{2k}}$. We are now interested the value of $a$, which can be simply calculated by $a = \frac{\mu(Y^{(k+1)})}{\mu(Y^{(k)})} = \frac{1+(\frac{2+\lambda}{1+\lambda})^{2k}}{1+(\frac{2+\lambda}{1+\lambda})^{2k+2}}$. It is easy to see from the previous expression that $a = O(\min(1, (\frac{1+\lambda}{2+\lambda})^2)) = O(\min(1, \frac{|\lambda|}{1+|\lambda|})$. In particular, we notice that in the region where the model suffers from rank collapse in the limit of infinite depth (i.e. $\lambda > -\frac{3}{2}$), we have that $a = O(\frac{|\lambda|}{1+|\lambda|})$. Note that this corresponds to choosing $|\lambda| = \Omega(\frac{a}{1-a})$, hence concluding the proof. $\qquad\square$

## A.13 ADDITIONAL EXPERIMENTS

### A.13.1 COMPARISON OF RANK COLLAPSE FOR THE S4 ARCHITECTURE WITH AND WITHOUT SKIP CONNECTION AND LAYERNORM

In the following, we consider a different SSM model, i.e. the S4 architecture Gu et al. (2022b), to analyze the effect of skip connections and LayerNorm on a different model. In particular, we use the S4 variant with diagonal $A$, i.e. S4D Gu et al. (2022a). We train the S4D architecture with 32 layers and 1.6 million parameters on the Cifar10 dataset. We then sample 32 images from this dataset and forward pass them through the trained model. Again, we calculate the normalized rank collapse measure $\mu$ at each layer. We repeat this for all considered cases, namely skip connections plus LayerNorm, only skip connections, only LayerNorm, and neither skip connections or LayerNorm.

Figure 4 shows that for S4D, even when removing the skip connection and the LayerNorm from the model, rank collapse is not observed. This an be explained by the fact that the weight matrices at different layers have a large Frobenius norm. From Theorem A.8, in order to have rank collapse, we must have that the weights matrices have Frobenius norm smaller than 1. Hence, the fact that we do not observe rank collapse does not violate our theory as in the case of weight matrices with large Frobenius norm, the upper bound presented in Theorem A.8 becomes vacuous. Additionally, note that while for Transformers and for Mamba-2 the observation of rank collapse depended (at least theoretically) on both the values of the weight matrices and the inputs values, for standard SSMs (e.g. S4) rank collapse only depends on the weight matrix. This renders the SSM model more "robust" to rank collapse, in the sense that to observe rank collapse one needs to find a precise parametrization of the model and cannot simply find an "adversarial" input sequence causing rank collapse (which instead could be done for every parametrization of Transformers and selective SSMs).

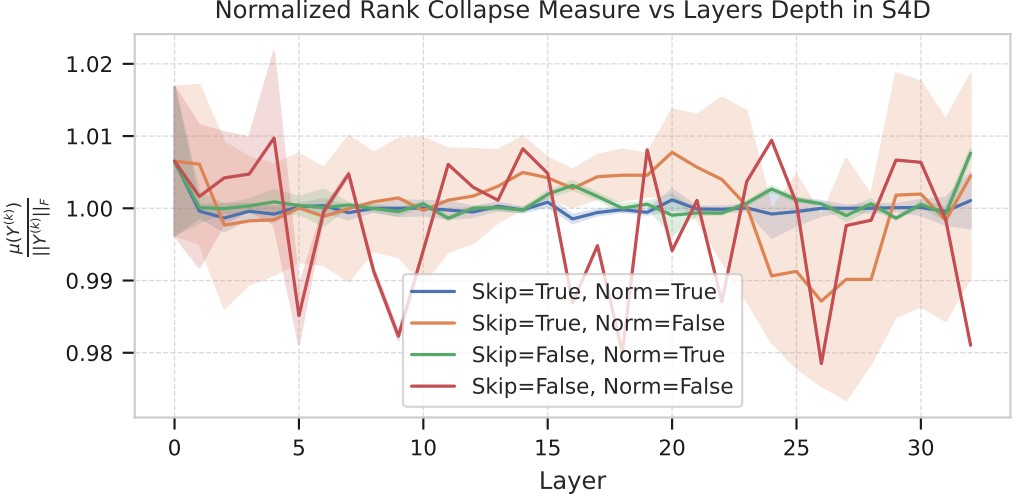

Figure 4: (Normalized) Rank Collapse measure as a function of layer depth for the S4D model. The shaded areas represent one standard deviation from the mean calculated over the 32 examples. Skip=True/False indicates we use a S4D architecture with/without skip connection whereas Layer-Norm=True/False indicate we use a S4D architecture with/without LayerNorm

### A.13.2 COMPARISON OF PERFORMANCE ON IMAGE LRA FOR DIFFERENT VALUES OF SKIP STRENGTH

In the following, we test and report the performance for different values of $\lambda$ for Transformers for the Image LRA task. In particular, we report a table with the performance of Transformers (with 8 layers) trained from scratch on the Image LRA dataset for different values of $\lambda$, namely -2, -1, 0, 1, 2. From the results, we can make the following conclusions: we can clearly see that rank collapse is happening for the case where $\lambda = 0$, proving the importance of having skip connections in the architecture. Indeed, the model for this value of $\lambda$ performs random guessing, meaning that it is not able to build useful representations of the inputs due to rank collapse. For the other cases, the performance is comparable across different values of $\lambda$.

| Model | $\lambda = -2$ | $\lambda = -1$ | $\lambda = 0$ | $\lambda = 1$ | $\lambda = 2$ |
|---|---|---|---|---|---|
| Transformers | 38.61 | 35.89 | 10.00 | 40.22 | 38.90 |

Figure 5: Performance metrics of Transformer and Mamba-2 for different values of $\lambda$.

### A.14 EXPERIMENTAL DETAILS

For both the LRA image task and the MQAR task we use the standard code bases provided online.[3] We use the standard transformer architecture, the linear attention model proposed in Katharopoulos et al. (2020), the Mamba architecture (Gu and Dao, 2024), and the Mamba-2 architecture (Dao and Gu, 2024). For the MQAR experiments we use the following training protocol:

- **Optimizer and schedule:** Weight decay of 0.1, linear warmup with duration of 10%, AdamW optimizer (Loshchilov and Hutter, 2019). For each run, we sweep the learning rates in `np.logspace(−4, −2, 4)` and train for 64 epochs. This is the same setup as in (Arora et al., 2023).

- **Initialization:** For all models we use their standard initialization and initialize $\lambda = -1$ in each layer.

---
[3]`https://github.com/HazyResearch/zoology`
and `https://github.com/google-research/long-range-arena`

- **Training duration:** We use a global batch size of $64$.

- **Width and depth:** For all runs, we use two layers (each with a sequence model and a MLP, interleaved with layer normalization). The model dimensions $d = 128$, state dimension $n = 64$, sequence length $L = 512$, and number of KV pairs $= 64$ are kept constant for all four architectures.

- **Position information:** Positional embeddings (Brown et al., 2020) are used for the attention architectures, but not for the SSM architecture classes. This is the same setup as in (Arora et al., 2023).

- **Data:** Each model is trained on 100,000 datapoints and evaluated on 3,000 datapoints. The data and its order are constant for all runs. This is the same setup as in (Arora et al., 2023).

For the LRA image experiments we use the following training protocol:

- **Optimizer and schedule:** Linear warmup with duration of 10%, AdamW optimizer (Loshchilov and Hutter, 2019). For attention-based models we use weight decay $0.0$ and learning rate $5e - 4$ and for SSM-based models we use weight decay $0.01$ and learning rate $2e - 4$.

- **Initialization:** For all models we use their standard initialization and initialize $\lambda = -1$ in each layer.

- **Training duration:** We use a global batch size of $64$.

- **Width and depth:** For all runs, we use four layers (each with a sequence model and a MLP, interleaved with layer normalization). The model dimensions $d = 256$ and state dimension $n = 64$ are kept constant for all four architectures.

- **Data:** Each model is trained on 35,200 datapoints and evaluated on 7,850 datapoints. The data and its order are constant for all runs.

Since we keep the model sizes constant among all architectures and do not optimize the hyperparameters, the accuracies reported in Table 1 are generally lower than in the literature. However, the main point of these experiments is to investigate if learning $\lambda$ affects performance.

