# OpenReview forum: "Lambda-Skip Connections: the architectural component that prevents Rank Collapse"
_ICLR.cc/2025/Conference — ICLR 2025 Poster_

### Official Review · Reviewer_LT9i · 2024-11-02

**Soundness:** 3
**Presentation:** 3
**Contribution:** 2
**Rating:** 5
**Confidence:** 4

**Summary:**

This paper addresses rank collapse, a phenomenon where embedding vectors in deep learning models converge to a uniform state. Building on previous studies that focused on transformers, this paper extends the analysis to State Space Models (SSMs). The study employs theoretical and empirical analysis to demonstrate how lambda-skip connections, LayerNorm, and gating mechanisms contribute to both the stability and expressivity of transformers and SSMs.

**Strengths:**

S1. The paper tackles the problem of rank collapse, extending its analysis from transformers to SSMs.
S2. Through theoretical proofs, the paper demonstrates that lambda-skip connections prevent rank collapse, preserving model expressivity in both transformers and SSMs.
S3. Experimental results show that lambda-skip connections and other components enhance expressivity and stability across different model architectures.

**Weaknesses:**

W1. The definition of the residual term between Eq.(3) and Eq.(6) is inconsistent, with ambiguity around whether X or V serves as the residual term. This inconsistency impacts the theoretical derivations that follow and should be clarified to ensure precise interpretations. Additionally, certain symbols, such as D, are used in both the SSM and LayerNorm contexts but represent different meanings. Distinct notation would improve readability and reduce potential confusion.
W2. While the experiments generally align with the theoretical predictions, some disparities remain unaddressed. For example, the theoretical threshold for λ appears more conservative than the empirical results suggest, and additional clarification would help. Further, the appendix notes rank stability even without skip connections, which might challenge the presented theory.
W3. The paper primarily focuses on rank collapse within the model’s architecture but does not connect this phenomenon to downstream task performance. Adding experimental results that measure downstream task performance in relation to model depth and skip connection strength could provide a more comprehensive assessment.

**Questions:**

Q1. Between Eq. (3) and Eq. (6), there is ambiguity regarding the residual term, specifically whether X or V serves as the residual component. This inconsistency could impact the theoretical derivations that follow. Could the authors clarify this definition? Additionally, using the same symbol D for both SSM and LayerNorm contexts creates potential confusion. Distinct notations would enhance clarity.
Q2. The theoretical conditions for λ appear to be conservative compared to empirical findings. Could the authors explain this discrepancy? Furthermore, the appendix notes cases of rank stability without skip connections, which might challenge the theory. An analysis of these cases would be valuable.
Q3. Could the authors provide additional experiments showing the model’s downstream performance as a function of layer depth and skip strength? Also, would the inclusion of alternative metrics, such as effective rank, offer a more comprehensive assessment of rank collapse?

---

> ### Author Response · Authors · 2024-11-19
>
> Thank you very much for the helpful feedback and pointers! We have reported the suggested changes in the updated version of the paper. In what follows we provide a further discussion on the points that were raised in the review:
>
> ***Question 1***. We very much thank the reviewer for pointing this out. The correct definition is the one in Equation 3, whereas the correct formula for Equation 6 is $Y^{(k+1)} = D^{(k)}(M^{(k)}V^{(k)}+\lambda Y^{(k)})$. We have corrected Theorem 4.1 and Equation 6 according to this change, the final result still holds with only minor corrections in the constants in the final bound and assumption. We reported the corrected version of the Theorem in the updated version of the paper. The corrected formulas are marked in blue. In particular, the following changes have been made: in the definition of the variable $b$ in Section 4.1., the denominator is $2 \lambda N d S C_M$ instead of $2 \lambda N S^2 C_M$ and the denominator becomes $\lambda^2-a(SC_M+|\lambda|)^2$ instead of $\lambda^2c^2-aS^2(C_M+|\lambda|^2)$. Equation 7 follows the change in the denominator of $b$. We apologize for the confusion arising from using the variable $D$ for both SSM and LayerNorm and we agree with the reviewer that this was not the best choice of variables. We have changed this in the updated version of the paper, introducing different variables for the SSM and LayerNorm to improve clarity.
>
> ***Question 2***. We thank the reviewer for raising this point. The main reason for the discrepancy between our theoretical predictions in the lower bound of Theorem 4.1 and the experiments is that the guarantee we propose is a worst case bound. This means that it takes into account all the possible input sequences that satisfy the assumptions stated in the Theorem. To clarify more the tightness and stringency of our bound, we have added in the updated version of the paper (in Section 4.2.3) an analysis of the tightness of the lower bound. In particular, we find an architecture and an input sequence for which, up to constants (in particular we study the interplay between $a$ and $\lambda$, considering $C_M$ and $S$ as constants), the lower bound is tight. One could see this as an adversarial sequence for which the smallest value of $\lambda$ to avoid rank collapse in the finite layer setting is the one proposed in our Theorem. However, it is likely that for “common”/arbitrary sequences (i.e. not chosen adversarially), the value of $\lambda$ to avoid rank collapse might be much lower, i.e. on the “average case” the lower bound is less tight, thus explaining the observed discrepancy. On the other hand, without any further assumptions on the input sequences or additional properties of the architecture, it is not possible to go beyond this bound to obtain a guarantee on rank collapse not occurring. Nevertheless, we think that extending this theory to study “average case” scenarios instead of worst cases is an interesting line of research and we plan to explore this in future work.
>
> The fact that in Figure 4 in the appendix we do not observe rank collapse even in the absence of skip connection can be explained by the fact that the weight matrices at different layers have a large Frobenius norm. From Theorem A.8, in order to have rank collapse, we must have that the weights matrices have Frobenius norm smaller than 1. Hence, the fact that we do not observe rank collapse does not violate our theory as in the case of weight matrices with large Frobenius norm, the upper bound presented in Theorem A.8 becomes vacuous. Additionally, note that while for Transformers and for Mamba-2 the observation of rank collapse depended (at least theoretically) on both the values of the weight matrices and the inputs values, for standard SSMs (e.g. S4) rank collapse only depends on the weight matrix. This renders the SSM model more "robust" to rank collapse, in the sense that to observe rank collapse one needs to find a precise parametrization of the model and cannot simply find an "adversarial" input sequence causing rank collapse (which instead could be done for every parametrization of Transformers and selective SSMs). We have expanded our discussion of Figure 4 in the appendix by including this remark and a better explanation of why we observe this.

---

> > ### Author Response · Authors · 2024-11-19
> > **Official Comment (Continued)**
> >
> > ***Question 3***. Although testing for classical downstream tasks is not possible to us due to time and computational constraints, since it would require to pre-train a new model from scratch using different values of $\lambda$, we test and report in the following the performance for different values of $\lambda$ for Transformers for the Image LRA task. In particular, we report a table with the performance of Transformers (with 8 layers) trained from scratch on the Image LRA dataset for different values of $\lambda$, namely -2, -1, 0, 1, 2. From the results, we can make the following conclusions: we can clearly see that rank collapse is happening for the case where $\lambda=0$, proving the importance of having skip connections in the architecture. Indeed, the model for this value of $\lambda$ performs random guessing, meaning that it is not able to build useful representations of the inputs due to rank collapse. For the other cases, the performance is comparable across different values of $\lambda$. We have added the table in the Additional Experiments section in the Appendix in the updated version of the paper.
> >
> >
> > | Model            | $\lambda=-2$ | $\lambda=-1$ | $\lambda=0$ | $\lambda=1$ | $\lambda=2$ |
> > |------------------|--------------|--------------|-------------|-------------|-------------|
> > | Transformer      | 38.61       | 35.89        | 10.00       | 40.22       | 38.90       |
> >
> >
> >
> >
> > In terms of alternative metrics, there are various ways to assess how “close” a matrix is to being rank 1, each focusing on how "distant" the matrix is from satisfying specific rank-related properties. For the metric we use, we evaluate how close the matrix's columns are to being equal to their average—a condition that holds if and only if the matrix is rank 1. Alternatively, the effective rank could be used to measure this distance. When considering rank-1 proximity, the effective rank evaluates how close the matrix is to having all but one singular value equal to zero. The effective rank is precisely 1 if and only if only one singular value is non-zero, which corresponds to the matrix being rank 1. The choice of metric is not unique, and effective rank is a viable alternative. However, we opted for our metric because previous works in the rank collapse domain ([1], [2]) used the same approach. This consistency facilitates a direct comparison of our results with those in prior studies. For results in the limit of infinite-depth networks, such as Theorem 4.3, we expect the effective rank to provide similar guarantees. This is because we show that, under appropriate conditions, the rank of the output layer converges to 1 in the limit. However, the rate of convergence for effective rank decay may differ from that of our chosen metric. For finite-layer results like Theorem 4.1, exploring the relationship between our metric and others, such as effective rank, would be a compelling direction for future work. This could clarify how our results translate across different metrics and deepen our understanding of their implications. We have added a brief remark on this in the updated version of the paper under Future Work.
> >
> > ***References***:
> >
> > [1] Y. Dong, J.-B. Cordonnier, and A. Loukas. Attention is not all you need: Pure attention loses rank doubly exponentially with depth, 2023. URL https://arxiv.org/abs/2103.03404.
> >
> > [2] X. Wu, A. Ajorlou, Y. Wang, S. Jegelka, and A. Jadbabaie. On the role of attention masks and layernorm in transformers, 2024a. URL https://arxiv.org/abs/2405.18781

---

> > > ### Author Response · Authors · 2024-11-27
> > > **Addressing any remaining concerns during the extended timeline**
> > >
> > > Dear Reviewer LT9i,
> > >
> > > Thank you once again for your insightful feedback.
> > >
> > > With the Discussion Phase extended until December 2, we would greatly appreciate it if you could let us know if there are any additional aspects we could address to further improve our paper. If our responses so far have not fully resolved any remaining concerns, please don’t hesitate to let us know, and we would be happy to work on addressing them during the extended timeline.
> > >
> > > Thank you again for your time and thoughtful comments!
> > >
> > > Best regards,
> > > Authors

---

> > > > ### Author Response · Authors · 2024-12-01
> > > > **Reminder of Deadline of Discussion Period**
> > > >
> > > > Dear Reviewer LT9i,
> > > >
> > > > We sincerely appreciate the time you have taken to provide feedback on our work, which has helped us to improve its clarity and correctness. This is a gentle reminder that the discussion phase will end in less than 2 days from this comment. We are happy to answer any further questions or concerns you may have before then,.
> > > >
> > > > If you agree that our responses to your reviews have addressed the concerns you listed, we kindly ask that you consider whether raising your score would more accurately reflect your updated evaluation of our paper. Thank you again for your time and thoughtful comments!
> > > >
> > > > Best regards,
> > > > Authors

---

### Official Review · Reviewer_jNQJ · 2024-11-04

**Soundness:** 3
**Presentation:** 3
**Contribution:** 3
**Rating:** 8
**Confidence:** 4

**Summary:**

This paper examines the phenomenon of rank collapse in general sequence model architectures, including transformers and state space models. To mitigate this issue, the paper proposes a parameterized version of the skip connection that multiplies the residual stream by a constant factor. Theoretical analysis identifies the conditions on the parameter sufficient to prevent rank collapse, and an analytical example demonstrates that neither the absence of skip connections nor the standard implementation prevents rank collapse. Finally, empirical evaluations support the findings of theoretical analysis.

**Strengths:**

This paper addresses the significant issue of rank collapse in sequence model architectures. It offers both theoretical analysis and empirical evaluation to support the proposed architectural component aimed at resolving this problem. I like the remark that provides the parameters corresponding to the practical architectural settings.

Additionally, the theoretical development and overall presentation of the paper are commendably clear and well-structured.

**Weaknesses:**

The theory investigates the sufficient conditions for preventing rank collapse in the worst-case scenario. This could imply that the required conditions are overly stringent.

**Questions:**

The rank collapse metric is not normalized in the definition. Would it be enough to lower bound the rank collapse metric, when the norm itself evolves across layers?

---

> ### Author Response · Authors · 2024-11-19
>
> We thank the reviewer for the positive feedback and comments! We really appreciate that they liked the paper and found the issue or rank collapse which we target significant. In the following, we address the mentioned weaknesses and the question raised:
>
> **Overly stringent condition**: The lower bound we consider in Theorem 4.1. is indeed a worst case bound and takes into account all the possible input sequences that satisfy the assumptions stated in the Theorem. To clarify more the tightness and stringency of our bound, we have added in the updated version of the paper (in Section 4.2.3) an analysis of the tightness of the lower bound. In particular, we find an architecture and an input sequence for which, up to constants (in particular we study the interplay between $a$ and $\lambda$, considering $C_M$ and $S$ as constants), the lower bound is tight. One could see this as an adversarial sequence for which the smallest value of $\lambda$ to avoid rank collapse in the finite layer setting is the one proposed in our Theorem. However, it is likely, as the reviewer suggests, that for “common”/arbitrary sequences (i.e. not chosen adversarially), the value of $\lambda$ to avoid rank collapse might be much lower, i.e. on the “average case” the lower bound is less tight, thus explaining the observed discrepancy. On the other hand, without any further assumptions on the input sequences or additional properties of the architecture, it is not possible to go beyond this bound to obtain a guarantee on rank collapse not occurring. Nevertheless, we agree with the reviewer that extending this theory to study “average case” scenarios instead of worst cases is an interesting line of research and we plan to explore this in future work.
>
> **Normalization of rank collapse metric**: We thank the reviewer for the interesting question. By normalized metric, we assume the reviewer refers to something like $\tilde{\mu}^{(k)} = \frac{\mu^{(k)}}{||Y^{(k)}||_F}$ (please let us know if this is not the case and what other type of normalization they had in mind, we will be happy to discuss this further). First of all, we observe that our result in Theorem 4.1. considers an architecture with LayerNorm. This means that  we have that $||Y^{(k)}||_F=N$ for all $k$, i.e. the Frobenius norm of the output of the k-th layer is always $N$. Hence, considering the normalized version of the metric leads to exactly the same result of the un-normalized one we use in this case (only scaled by a constant $N$). Hence, studying these two versions of the metric would lead to the exact same conclusions for the setting of Theorem 4.1. However, it would be interesting to study the normalized version of the metric in cases where no LayerNorm is present in the architecture, as it is likely that under these circumstances the two metrics could present different behavior and dynamics. Finally, we conclude by saying that our choice of using the un-normalized metric was driven by the fact that previous works in the literature ([1], [2]) use this version and hence studying this metric made comparing our and previous findings easier. Please let us know if this addresses the question raised, or if we have misunderstood the question.
>
> **References**:
>
> [1] Y. Dong, J.-B. Cordonnier, and A. Loukas. Attention is not all you need: Pure attention loses rank doubly exponentially with depth, 2023. URL https://arxiv.org/abs/2103.03404.
>
> [2] X. Wu, A. Ajorlou, Y. Wang, S. Jegelka, and A. Jadbabaie. On the role of attention masks and layernorm in transformers, 2024a. URL https://arxiv.org/abs/2405.18781

---

> > ### Comment · Reviewer_jNQJ · 2024-11-26
> >
> > Thank you for the response to my review. The response addresses the issues and questions that I raised. I keep my current score unchanged.

---

> > > ### Author Response · Authors · 2024-11-26
> > >
> > > We thank the reviewer for reading our rebuttal and we are happy to hear our answer addressed all your concerns!

---

### Official Review · Reviewer_Av2c · 2024-11-04

**Soundness:** 3
**Presentation:** 2
**Contribution:** 2
**Rating:** 6
**Confidence:** 3

**Summary:**

This paper analyzes the rank collapse of SSM due to identical $\lambda$ skip connections. The authors provide a rigorous convergence rate for the rank collapse and offer sufficient guarantees to prevent it. Experimental results demonstrate the effectiveness of their analysis.

**Strengths:**

1. The lower boundary of the rank collapse of $\lambda$ skip connections is analytically derived. The results agree well with empirical analysis.
2. The paper presents the convergence rate in the absence of skip connections, contributing valuable insights.

**Weaknesses:**

1. The authors analyze the $\lambda$ skip connections. However, the skip strength $\lambda_k$ may vary on different layers.The paper should discuss how the findings hold up under these varying conditions. Additionally, many models implement skip connections selectively across layers rather than uniformly. A discussion on the generalizability of the results would enhance the paper.
2. Theorem 4.1 paves the way to choose suitable $\lambda$. However, in Figure 2, it appears that when $\lambda$ is sufficiently large, the rank collapse index shows little variation. Clarification on how to determine the optimal value of $\lambda$ would be beneficial.
3. Based on theorem 4.1, could the authors explore adding constraints to the parameters to optimize $C_M$, $S$ and $c$ for improved neural network performance?

**Questions:**

See above.

---

> ### Author Response · Authors · 2024-11-19
>
> Thank you for the insightful comments! We address in the following the points that were raised:
>
> **Skip connection strength**. This is a really good question and we thank the reviewer for having raised this point. In the case where we let all the layers have different values of the skip strength $\lambda_k$ (which for instance would be the case if we make the parameter $\lambda$ learnable), we can generalize the result in Theorem 4.1. by making the following small modification: in order to still satisfy the condition $\mu(Y^{(k)}) \geq a \mu(Y^{(k-1)})$, we can simply adapt Equation 7 to be $\lambda_k^2-a(S_kC_{M_k}+|\lambda_k|)^2>0$, where $S_k = ||C^{(k)}_V||_F and C_{M_k} = ||M^{(k)}||_F. Furthermore, the fact that we can choose $\lambda_k$, means that we can also choose different values of $a$ in different layers. Consider the following example: suppose we aim to ensure that $\mu(Y^{(2)}) \geq 0.25 \mu(Y^{(0)})$. According to the setting of Theorem 4.1, the only way to achieve this is by selecting $a=0.5$ and adjusting $\lambda$ to satisfy the condition in Equation 7 for this specific value of $a$. However, if we allow $\lambda_k$ to vary across layers, we gain additional flexibility. For instance, in the first layer, we could choose a $\lambda_k$ that does not satisfy the condition in Equation 7 for $a = 0.5$ (and thus cannot guarantee $\mu(Y^{(1)}) \geq 0.5 \mu(Y^{(0)})$), but instead satisfies the condition for a smaller value, for instance $a = 0.4$. In the second layer, we could then select a $\lambda_k$ that satisfies the condition in Equation 7 for $a = 0.625$. This setup still achieves the desired result: $\mu(Y^{(2)}) \geq 0.625 \times 0.4 \mu(Y^{(0)}) = 0.25 \mu(Y^{(0)})$.This demonstrates the added flexibility and robustness provided by allowing \(\lambda_k\) to vary across layers. However, when skip connections are not applied uniformly, greater caution is required to generalize the results, as specific lower bounds would need to be developed for cases without skip connections—an aspect we do not address in this work. That said, to the best of our knowledge, the vast majority of Transformer-based models, SSMs, and Mamba (the architectures considered in this paper) consistently apply skip connections uniformly across layers. Thus, our theoretical results remain applicable to these architectures. Nonetheless, we agree with the reviewer that extending our results to scenarios where skip connections are applied non-uniformly across layers would be an interesting direction for future research. We have addressed these important points at the end of Section 4.1. in the updated version of the paper.
>
> **Optimal value for $\lambda$**. This is a great question. Currently, there is not a definition of what it means for $\lambda$ to be optimal. Does the reviewer refer to something like the smallest $\lambda$ such that the rank collapse metric at the output layer is greater than some threshold? If this is the case,  the optimal $\lambda$ would be threshold dependent. Intuitively, the lower the threshold we want to achieve, the lower the optimal $\lambda$ will be (and vice-versa). One way to find this optimal $\lambda$ would be to use the lower bound proposed in Theorem 4.1: by tuning the value of $a$ we can control the threshold we want the metric to satisfy and then we can choose $\lambda$ accordingly to satisfy Equation 7 with the chosen value of $a$. However, as noted by other reviewers, the bound in some situations is too conservative and lower values of $\lambda$ are enough to guarantee that the rank collapse metric is above the desired threshold. Although in our work we do nor provide an explicit way for finding an optimal $\lambda$ (since we do not introduce any notion of optimality), another approach for finding good values of $\lambda$ is by making $\lambda$ a learnable parameter and provide good initialization for it (for example, for the experiments in Table 1, we found that $\lambda$= -1 was a good choice for initialization). Moreover, the small variation observed in Figure 2 for larger values of $\lambda$ can be explained using the lower bound proposed in Theorem 4.1. Specifically, as $|\lambda| \to \infty$, the parameter $a \to 1$. This implies that the rank collapse metric decreases much more slowly with the number of layers. In the limit, it does not decrease at all, remaining at or above the rank collapse measure at the input. We have included a brief discussion on this point in the updated version of the manuscript (see Section 5.1).

---

> > ### Author Response · Authors · 2024-11-19
> > **Official Comment (Continued)**
> >
> > **Other parameters**. If we understand correctly, the reviewer is suggesting adding constraints to the parameters of the neural network to control the value of $C_M$ and $S$. Adding constraints to the parameters of the neural network can come at two costs: first, adding constraints to the parameters would lead to very expensive training of the neural network (in particular, it requires to perform projections in the region we are constraining the parameters in). Second, constraining parameters might reduce models’ downstream performance rather than improving it since we might hinder the model from finding the minimizer of the loss function, which might lie outside the constrained region. Instead, we propose that making $\lambda$ learnable during training is a better way of dealing with this as it both solves the problem of stability and rank collapse and, at the same time, it still finds a $\lambda$ leading to comparable, if not superior performance than setting $\lambda=1$ (see again Table 1). Furthermore, in this work our aim is not analyze how to constraint the backbone architecture (i.e. the attention layer or the SSM block) to prevent rank collapse, which can be convoluted, but rather explore how we can prevent rank collapse by performing simpler modifications to sequence models’ architectures, by e.g. modifying the skip connection by making it a learnable parameter. Note that this only adds one learnable parameter per layer and hence it does not affect training costs, since every layer has hundreds of thousands or millions of parameters.

---

> > > ### Author Response · Authors · 2024-11-27
> > > **Addressing any remaining concerns during the extended timeline**
> > >
> > > Dear Reviewer Av2c,
> > >
> > > Thank you once again for your insightful feedback.
> > >
> > > With the Discussion Phase extended until December 2, we would greatly appreciate it if you could let us know if there are any additional aspects we could address to further improve our paper. If our responses so far have not fully resolved any remaining concerns, please don’t hesitate to let us know, and we would be happy to work on addressing them during the extended timeline.
> > >
> > > Thank you again for your time and thoughtful comments!
> > >
> > > Best regards,
> > > Authors

---

> > > > ### Comment · Reviewer_Av2c · 2024-11-28
> > > >
> > > > Thanks for your response, you mainly answer my questions. I keep my current score unchanged.

---

> > > > > ### Author Response · Authors · 2024-11-28
> > > > >
> > > > > We thank the reviewer for reading our rebuttal and we are happy to hear our answer addressed all your concerns!

---

### Official Review · Reviewer_gCrW · 2024-11-04

**Soundness:** 3
**Presentation:** 3
**Contribution:** 3
**Rating:** 6
**Confidence:** 2

**Summary:**

Dao and Gu [https://arxiv.org/pdf/2405.21060] established a form of equivalence between transformers and continuous-time state-space models.  In a different development, Dong et al. [https://arxiv.org/abs/2103.03404] showed that self-attention layers without skip connections or MLPs suffer from "rank collapse" ― with increasing layers, the output matrix tends to rank-1, i.e., all token positions tend to the same representation.

The present submissions puts these together to show that rank collapse is a problem also for state-space models.  It shows that the skip connection provides vital protection against rank collapse, but that a weighted addition (with weight $\lambda$ which may be regarded as a hyperparameter, or perhaps trainable) with the skip connection is more flexible.

**Strengths:**

Reasons to accept:
* Identifies rank collapse problem in state-space models like MAMBA, similar to earlier discovery of this problem in transformer-type networks.
* Identifies skip strength parameter $\lambda$ as an important knob to limit the damage of rank collapse.

**Weaknesses:**

Reasons to reject:
* Given the two papers on which this paper builds, it might be argued that the present work is relatively incremental.  (That being said, I appreciate the candor while setting up the contributions of this paper, and I learnt something from it.)

**Questions:**

I have a reasonable estimate of creativity and technical depth, but it is difficult for me to assess impact.  I am not familiar with the area and my assessment has limited confidence. I would not, for example, know if rank collapse is widely appreciated within even the transformer "community" (if there is any such thing).  I have not seen MAMBA become that visible or widely used compared to standard transformer-based LLMs, but cannot speculate if rank collapse played a role.  $\lambda$-tuning for robustness seems quite useful, but again I do not know the area well enough to know if, in practice, $\lambda=1$ is frequently dangerous.  If the authors point to specific places in the paper where the above issues are discussed, or add some more motivating support, that would be helpful.

A few writing style and notation nits:

L156-L159 set up $X^{(k)}$ as layer input and $Y^{(k)}$ as layer output.  However, equation (1) introduces $O^{(k)}$ without explaining it will provide skipping ability in equation (2).

L174-L178 There seem to be some inconsistent subscripts and superscripts. On one side we see $A^{(k)}_t, B^{(k)}_t$ etc. But just after the displayed equation, for LTI systems we see the superscript $(k)$ disappear, without an explanation if this is because the LTI system is assumed to have one layer.

L888-L903 While setting up expressions and bounds with so many variables, it helps to afterward highlight the most import 1-2 variables, and give qualitative connections between their typical values in practice and the implications on the bounds.  E.g., how easy or difficult would be to choose an acceptable $\lambda$ in a typical LLM?  Also, some of the definitions like $S$ are very far from the proofs in the appendix.

---

> ### Author Response · Authors · 2024-11-19
>
> Thank you for the insightful comments! We address in the following the points that were raised:
>
> **Novelty**: We thank the reviewer for raising this point. We will make sure to clarify the novelty of our contribution in the Introduction and Conclusion sections. While it is true that some of our findings—particularly those concerning the upper bounds of rank collapse for Mamba—share similarities with results on rank collapse in Transformers presented in the two foundational papers we reference, our primary theoretical contribution, namely the lower bound on the rank collapse measure in Theorem 4.1, is entirely original. To the best of our knowledge, this represents the first instance (even within the context of Transformers) of a general lower bound on rank collapse of this kind being proposed. Additionally, we offer for the first time a mechanistic solution to prevent rank collapse by introducing a skip strength parameter. This parameter, which can be viewed as a novel architectural feature, deviates from the standard practice of defaulting the skip strength to 1. We also highlight that the simplicity of this proposed novel mechanism makes our solution straightforward to implement.
>
> **Impac**t: We appreciate the points brought up by the reviewer. We will add clarifications on these points throughout the paper. We have broken down the impact concern into two parts: (i) why is rank collapse important?, (ii) why is rank collapse relevant in State Space Model architectures (such as Mamba)?, and (iii) what is the usefulness of $\lambda$-skip connections?
>
> (i) Rank collapse poses a significant challenge during both inference and training. When a model is susceptible to rank collapse, it results in tokens being mapped to nearly identical representations, severely limiting the model's expressivity and representational capacity ([1], [2], [3], [4]). Furthermore, previous studies [3] have demonstrated that rank collapse during training can lead to instabilities, primarily caused by vanishing gradient issues. Therefore, gaining a deeper understanding of this phenomenon—its origins, conditions for occurrence, and potential prevention strategies—is essential for designing models that are more robust, stable, and expressive.
>
> (ii) To the best of our knowledge, the Rank Collapse phenomenon is well-recognized within the Transformer community. For example, the original paper on rank collapse [5], which first highlighted the importance of skip connections in mitigating this issue, has garnered nearly 400 citations, and several subsequent studies have built upon its findings, as outlined in our Related Work section. While it is true that Mamba has not yet reached the level of prominence of Transformers—likely due to its relatively recent introduction less than a year ago [6]—it is beginning to gain traction across a variety of fields, such as biology [7]. Given this, we believe it is crucial to raise awareness within the SSM/Mamba community that rank collapse is not a phenomenon exclusive to Transformer architectures, but could also negatively impact the performance of recurrent-like architectures. In doing so, we hope our work will provide valuable insights for developing more robust SSM-based architectures, thereby expanding their potential applications.
>
> (iii) While the choice of $\lambda = 1$ might, in some circumstances, not result in rank collapse (often when combined with LayerNorm), allowing for a learnable $\lambda$ or treating it as a hyperparameter offers several advantages. First, it enhances the model's expressivity, leading to comparable and in some cases improved performance (as evidenced by our experiments in Table 1). We remark that this is possible with minimal computational overhead: adding just one additional learnable parameter per layer is negligible compared to the vast number of learnable parameters in Transformers or SSM layers, typically ranging from hundreds of thousands to millions in the smallest cases. Second, it improves the model's stability. Specifically, initializing $\lambda$ with a negative value (e.g., $\lambda$=−1, as in our experiments in Table 1) can be viewed through a control-theoretic lens as implementing a negative feedback loop. Negative feedback systems are generally more stable compared to positive feedback systems, which is effectively the case when $\lambda$=1, as is commonly used. Therefore, we argue that introducing a learnable $\lambda$ offers significant potential benefits—enhancing stability and expressivity—at a very low computational cost. This makes it a promising addition to traditional architectures. In this work we simply lay the foundation, and we believe that additional analysis on the strength of the skip connection could be done to prevent and control the behavior of foundation models from a control-theoretic standpoint.

---

> > ### Author Response · Authors · 2024-11-19
> > **Official Comment (Continued)**
> >
> > **Typos and Notation**: We very much thank the reviewer for bringing up these points.
> >
> > - We are sorry to hear L156-159 were not clear to the reviewer. Could the reviewer please clarify what they refer to  with “explaining $O^{(k)}$ provides skipping abilities in Equation 2”? We will be happy to address this concern and update the manuscript accordingly.
> >
> > - We apologize for the confusion and indeed there should be superscript (k) also in the expressions following L174-L178, as we are considering a deep LTI system with multiple layers. We have corrected this in the updated version of the paper.
> >
> > - We agree with the reviewer that the passage in lines L888-L903 is a bit convoluted as it involves many different variables, and we apologize for the confusion. We have addressed this in the updated version of the paper by reporting the definition of the variables (like $S$ and $C_M$) inside the proof as well. Furthermore, we have also added an explanation on the most important variables in the bound, their typical values, relationship and implications for the bound (in particular how to choose suitable values for $\lambda$) in the Appendix after the proof of Theorem 4.1. For completeness, we provide a brief explanation here as well: in the presented bound, the key variables of interest are $\lambda$, $a$ and $\mu(Y^{(k)})$. Specifically, we aim for $\mu(Y^{(k)})$ to be as far from 0 as possible, as values close to 0 indicate rank collapse. Furthermore, as we mentioned in lines 295-300, the ideal value of $a$ would be 1 (since this would guarantee that the rank collapse metric is non decreasing over layers), although in order to satisfy Equation 7 this value cannot be chosen. In practice, the typical value of $\lambda$ is 1. The key relationship between $a$ and $\lambda$ is the following: in order to guarantee values of $a$ closer and closer to 1 (and hence to ensure higher values of the rank collapse metric at the final layer) we must choose larger values for $|\lambda|$. Additionally, $N$ represents the input sequence length, which varies based on the task. For example, $N$ might be on the order of tens for simple question answering tasks, but it could scale to hundreds or thousands when summarizing a long document. $d$ instead represents the embedding dimension, typically in the order of tens or hundreds. Regarding the selection of $\lambda$, we propose two possible approaches: treating it as a hyperparameter or making it learnable. In the first approach, $\lambda$ would be chosen through a standard hyperparameter optimization procedure, testing different values and evaluating their impact on both performance and the rank collapse measure. While effective, this method requires multiple training runs to identify the best value. In contrast, the second approach—making $\lambda$ learnable—offers significant advantages. It automates the process of finding an optimal $\lambda$, eliminating the need for manual hyperparameter tuning and requiring only a single training run. This is both more efficient and practical. For these reasons, we adopted the learnable $\lambda$ approach in our experiments, as illustrated in Table 1.
> >
> >
> > **References**:
> >
> > [1] X. Wu, A. Ajorlou, Z. Wu, and A. Jadbabaie. Demystifying oversmoothing in attention-based graph neural networks, 2024b. URL https://arxiv.org/abs/2305.16102.
> >
> > [2] H. Daneshmand, J. Kohler, F. Bach, T. Hofmann, and A. Lucchi. Batch normalization provably avoids rank collapse for randomly initialised deep networks, 2020. URL https://arxiv.org/abs/2003.01652
> >
> > [3] L. Noci, S. Anagnostidis, L. Biggio, A. Orvieto, S. P. Singh, and A. Lucchi. Signal propagation in transformers: Theoretical perspectives and the role of rank collapse, 2022. URL https://arxiv.org/abs/2206.03126.
> >
> > [4] X. Wu, A. Ajorlou, Y. Wang, S. Jegelka, and A. Jadbabaie. On the role of attention masks and layernorm in transformers, 2024a. URL https://arxiv.org/abs/2405.18781.
> >
> > [5] Y. Dong, J.-B. Cordonnier, and A. Loukas. Attention is not all you need: Pure attention loses rank doubly exponentially with depth, 2023. URL https://arxiv.org/abs/2103.03404.
> >
> > [6] A. Gu and T. Dao. Mamba: Linear-time sequence modeling with selective state spaces, 2024. URL https://arxiv.org/abs/2312.00752.
> >
> > [7] Y. Schiff, C. Kao, A. Gokaslan, T. Dao, A. Gu and V. Kuleshov. Caduceus: Bi-Directional Equivariant Long-Range DNA Sequence Modeling. URL https://arxiv.org/abs/2403.03234

---

> ### Author Response · Authors · 2024-11-27
> **Addressing any remaining concerns during the extended timeline**
>
> Dear Reviewer gCrW,
>
> Thank you once again for your insightful feedback.
>
> With the Discussion Phase extended until December 2, we would greatly appreciate it if you could let us know if there are any additional aspects we could address to further improve our paper. If our responses so far have not fully resolved any remaining concerns, please don’t hesitate to let us know, and we would be happy to work on addressing them during the extended timeline.
>
> Thank you again for your time and thoughtful comments!
>
> Best regards,
> Authors

---

> > ### Author Response · Authors · 2024-12-01
> > **Reminder of Deadline for Discussion Period**
> >
> > Dear Reviewer gCrW,
> >
> > We sincerely appreciate the time you have taken to provide feedback on our work, which has helped us to improve its clarity. This is a gentle reminder that the discussion phase will end in less than 2 days from this comment. We are happy to answer any further questions or concerns you may have before then,.
> >
> > If you agree that our responses to your reviews have addressed the concerns you listed, we kindly ask that you consider whether raising your score would more accurately reflect your updated evaluation of our paper. Thank you again for your time and thoughtful comments!
> >
> > Best regards,
> > Authors

---

> > > ### Comment · Reviewer_gCrW · 2024-12-02
> > > **Many thanks for engaging in the rebuttal discussions**
> > >
> > > Your clarifications will greatly help improve the quality of the next version of the manuscript.
> > > Based on my less-than-ideal familiarity of the area and confidence, I wish to hold my overall score.
> > > The paper may be in a good position anyway. All the best!

---

> > > > ### Author Response · Authors · 2024-12-02
> > > >
> > > > We thank the reviewer for taking the time to read our rebuttal and we are happy to hear our answer addressed all your concerns!

---

### Author Response · Authors · 2024-11-19

We would like to thank all the reviewers for their time and effort evaluating our paper. We believe the insightful reviews helped us to greatly improve the paper. The main contents of the rebuttal are clarifications on the novelty and the impact of our work, a discussion on the tightness of our bound in Theorem 4.1. and how to choose $\lambda$ in practice and extended evaluations on the Image LRA benchmark for different values of $\lambda$. We have addressed all the concerns that have been raised by the reviewers and we hope they find our answers, modifications and explanations satisfactory. Furthermore, we have introduced the suggested changes in the updated version of the paper. The reviewers can find the newly introduced or modified parts highlighted in blue.

**Novelty and Impact**: In response to the reviews, we clarified our theoretical contribution, its novelty and the practical importance and impact of rank collapse. To the best of our knowledge, Theorem 4.1. represents the first instance of a general lower bound on rank collapse being proposed. Additionally, we offer for the first time a mechanistic solution to prevent rank collapse by introducing a skip strength parameter, which can be seen as a new architectural component. Models suffering from rank collapse have both limited inference and training capabilities, caused by the reduced representational capacity. Hence, since State Space Models and Mamba in particular are gaining significant traction and rank collapse has not yet been studied for these models, we think it is of great importance to shed light on when this phenomenon arises and how it can be prevented. Finally, the introduction of a learnable parameter $\lambda$, while it does not influence the cost of training, improves models’ expressivity and stability.

**Tightness of lower bound**: Reviewers expressed some concerns relatively to the fact that there is a discrepancy between the lower bound we presented in Theorem 4.1. and experiments. To address this, we added a new paragraph (section 4.2.3) in the updated version of the paper, where we discuss and show our lower bound is tight. The discrepancy with experiments can then be explained by the fact that the lower bound is a worst-case guarantee and hence it accounts for all possible input sequences. On the other hand, it is likely the case that for most input sequences much lower values of $\lambda$ are enough to prevent rank collapse from happening, as we showed in the experimental section.

**Additional experiments**: In response to the reviews, we performed additional experiments exploring the relationship between different values of $\lambda$ and the performance of models on the Image LRA task. In particular, we train Transformers for 5 different values of $\lambda$ from scratch on the task. We then report the final performance for the different values of $\lambda$. We note that for $\lambda=0$ the model performs random guessing, hinting that in this case the model suffers from rank collapse and hence showing the importance of including skip connections in the architecture. The models with the remaining values of $\lambda$ instead achieve similar performance.

**Finding the optimal $\lambda$**: In response to the reviewers, we clarified both how we can select acceptable values of $\lambda$ (i.e. satisfying the bound in Equation 7) and optimal values of $\lambda$. Although a concrete and precise notion of optimality would need to be defined to discuss this rigorously, in both cases there are two ways in which we can address these two tasks: the first is by letting $\lambda$ to be a hyperparameter whereas the second is to make $\lambda$ learnable. We think that the second option is more efficient and practical, since it automates the search for an optimal $\lambda$, eliminating the need for manual hyperparameter tuning and requiring only a single training run.

**Normalization and choice of rank collapse metric**: In response to reviews, we address the questions related to our choice of rank collapse metric and in particular the possibility of using a normalized version of the metric or other metrics (such as effective rank) to assess rank collapse. First, we clarify that our choice of metric was driven by the fact that this metric was used in previous papers addressing rank collapse. Hence, employing this metric enabled us to compare our findings with previous results. Furthermore, we think that the use of a normalized metric would be interesting but this metric would still be equal to ours in settings where LayerNorm is deployed.

**Typos and notation**: In response to the reviews, we addressed their concerns about typos and notation individually. We would like to bring to the attention of all reviewers the changes introduced in Equation 6, which lead to slight modifications of some constants in Theorem 4.1. and Equation 7.

---

### Author Response · Authors · 2024-11-25
**Any additional questions from the Reviewers on our Rebuttal?**

Dear Reviewers,

We would like to thank you again for your thoughtful reviews and valuable feedback. As the rebuttal period is coming to an end, we would appreciate it if you could let us know if our responses have addressed your concerns and whether you still have any additional questions about our rebuttal.
We would be happy to do any follow-up discussion or address any additional comments.

Best regards,
The Authors

---

### Meta-Review · Area_Chair_Bh57 · 2024-12-19

**Metareview:**

The paper discusses the phenomenon of rank collapse in LLMs, extending the study done on transformers to the SSM architecture. This issue is highly motivated given the popularity of SSM-based LLMs. The reviews note that the paper provides valuable insights to the problem coupled with a theoretically backed solution to it. This solution is mentioned to be practical and shown to be effective with convincing empirical evidence.

The reviews did mention some vague parts in the paper requiring clarifications and suggested ways in which the analysis can be made more thorough. From the discussion, these issues are either resolved by the authors explanations or appear to be minor. The changes required to integrate the needed parts of the discussion into the paper are quite minor and can be made towards a camera-ready version.

I believe the paper’s strengths outweigh the relatively minor weaknesses pointed out in the reviews, leading me to recommend accepting it to ICLR.

**Additional Comments On Reviewer Discussion:**

The discussion did not end with a consensus, and one reviewer (LT9i) remained leaning towards rejecting the paper. This being said, considering the strengths mentioned in the other reviews, the mild nature of the mentioned weakness, and the convincing reply of the authors, I still believe the paper meets the bar for ICLR.

---

### Decision · Program_Chairs · 2025-01-22

Accept (Poster)